# Operative dimensions in unconstrained connectivity of recurrent neural networks

**Renate Krause, Matthew Cook, Sepp Kollmorgen, Valerio Mante**[*] **and Giacomo Indiveri**[*]

Institute of Neuroinformatics, University of Zurich and ETH Zurich, Zurich, Switzerland
Neuroscience Center Zurich, University of Zurich and ETH Zurich, Zurich, Switzerland
Corresponding authors: rekrau@ini.uzh.ch, valerio@ini.uzh.ch
[*]Equal contribution

## Abstract

Recurrent Neural Networks (RNNs) are commonly used models to study neural computation. However, a comprehensive understanding of how dynamics in RNNs emerge from the underlying connectivity is largely lacking. Previous work derived such an understanding for RNNs fulfilling very specific constraints on their connectivity, but it is unclear whether the resulting insights apply more generally. Here we study how network dynamics are related to network connectivity in RNNs trained without any specific constraints on several tasks previously employed in neuroscience. Despite the apparent high-dimensional connectivity of these RNNs, we show that a low-dimensional, functionally relevant subspace of the weight matrix can be found through the identification of *operative* dimensions, which we define as components of the connectivity whose removal has a large influence on local RNN dynamics. We find that a weight matrix built from only a few operative dimensions is sufficient for the RNNs to operate with the original performance, implying that much of the high-dimensional structure of the trained connectivity is functionally irrelevant. The existence of a low-dimensional, operative subspace in the weight matrix simplifies the challenge of linking connectivity to network dynamics and suggests that independent network functions may be placed in specific, separate subspaces of the weight matrix to avoid catastrophic forgetting in continual learning.

## 1 Introduction

A central goal in neuroscience is to understand how groups of tightly interconnected neurons generate the complex network dynamics that underlies behavior. To this end, ever larger experimental datasets on neural anatomy, neural activity and the corresponding behavior are collected and analyzed and new experimental tools to extend the amount and quality of such data are continuously developed. However, in most settings it remains an open question how the underlying neural connectivity is able to generate the observed neural dynamics. Progress in how to inspect and interpret these complex datasets, and in particular on the relation between neural structure and function, may come in particular from new theoretical frameworks [1, 2].

Artificial Recurrent Neural Networks (RNNs) are a promising tool to develop such theoretical frameworks in a well-controlled and flexible setting [3, 4, 5]. Previous work on RNNs explained how a specifically designed connectivity can give rise to desired network dynamics [6, 7, 8]. Further theoretical work on RNNs with random, recurrent weights provided detailed insights into the properties of neural dynamics emerging from largely unstructured connectivity [9, 10, 11, 12, 13]. More recent work related structure to function in feedforward networks [14, 15] or RNNs with specific connectivity constraints. For example, network motifs in threshold-linear networks are used to predict

the existence of fixed points of the dynamics [16]. Similarly, a principled understanding of dynamics and the role of different cell classes in computations can be achieved for RNNs with low-dimensional weight matrices (low-rank RNN [17, 18]). At present, it remains unclear if and how the resulting findings can be generalized to RNNs which are not subject to such constraints.

In this work, we study how the network dynamics are related to the network connectivity in vanilla RNNs that are trained using a gradient-based approach without imposing any specific constraints on the network weight matrix. We find that the weight matrix of the trained RNN is consistently high-dimensional, even when trained on tasks resulting in dynamics that are low-dimensional. Notably, we are nonetheless able to identify a low-dimensional subspace within the high-dimensional weight matrix that is sufficient to perform the trained task. We identify this functionally relevant subspace of the connectivity through the definition of a set of *operative dimensions*, which we define as components of the network connectivity that have a large impact on computationally relevant local dynamics produced by the network.

This ability to identify functionally relevant subspaces in weight matrices improves our understanding of how the network connectivity generates the observed network dynamics and thereby makes RNNs into a more interpretable model for neuroscience and machine learning applications.

## 2   Results

We perform our analyses on vanilla RNNs trained without regularization terms, using the standard RNN equation:

$$\tau \dot{\mathbf{x}}_t = -\mathbf{x}_t + \mathbf{W}\mathbf{r}_t + \mathbf{B}\mathbf{u}_t + \boldsymbol{\sigma}_t \tag{1}$$

where $\mathbf{x}_t \in \mathbb{R}^N$ are the linear activities of the $N$ hidden units over time $t$ with $\mathbf{r}_t = tanh(\mathbf{x}_t)$, $\mathbf{W} \in \mathbb{R}^{N \times N}$ is the recurrent weight matrix of the hidden units and $\tau \in \mathbb{R}$ is the time constant ($\tau = 10\,ms$, $dt = 1ms$). We consider RNNs of $N = 100$ noisy units, where each element of $\boldsymbol{\sigma}_t$ is drawn from a Gaussian distribution $\mathcal{N}(\mu = 0, \sigma = 3.1623\sqrt{dt} \approx 0.1)$. The network output is defined as:

$$\mathbf{z}_t = \mathbf{Y}\mathbf{r}_t \tag{2}$$

with output readout weights $\mathbf{Y} \in \mathbb{R}^{Z \times N}$. The task-dependent, time-varying inputs $\mathbf{u}_t \in \mathbb{R}^U$ are projected onto the hidden units with input weights $\mathbf{B} \in \mathbb{R}^{N \times U}$. Note that only the inputs vary across different conditions within a task. For any given condition, the cost is defined as:

$$cost = \frac{1}{ZT} \Sigma_{i=1}^Z \Sigma_{t=1}^T (z_t^*(i) - z_t(i))^2 \tag{3}$$

where $z_t^*(i)$ is the desired output. All network weights ($\mathbf{B}$, $\mathbf{W}$, $\mathbf{Y}$) are randomly initialized, and networks are trained to minimize the summed costs across all conditions.

The RNNs are trained separately on two previously proposed tasks: context-dependent integration [19] and sine wave generation [20] (see appendix section A.3.8 for additional results on sequential MNIST). The dynamics of RNNs trained on these tasks is well understood, and was shown to be largely independent of the type of employed RNN [21]. In context-dependent integration, the RNN receives two noisy, sensory inputs (between -1 and 1) and two static, context inputs (0 or 1). The network is trained to select one of the two sensory inputs (depending on the currently active context input; i.e. select input sensory$_i$ in context$_i$) and integrate it over time (Fig. 1a, b). The network should reach choice$_1$ or choice$_2$ if the average of the contextually relevant sensory input is positive or negative, respectively. In the sine wave generation task, the RNN receives one static input and is trained to output a sine wave whose target frequency is given by the level of this static input (Fig. 1f, g; for details on task structures see section A.1.1). For both tasks we trained 20 RNNs (different random initial connectivity) with gradient-based optimization to minimize the cost (Eq. 3; for details see section A.1.2).

### 2.1   High-variance dimensions

To characterize the relation between network connectivity and dynamics, we first consider the functional relevance of *high-variance* dimensions (as in analyses of low-rank RNN [17]) defined as the left singular vectors of the RNN weight matrix $\mathbf{W}$:

$$\mathbf{W} = \Sigma_{i=1}^N \mathbf{w}_i s_i \mathbf{v}_i^T \tag{4}$$

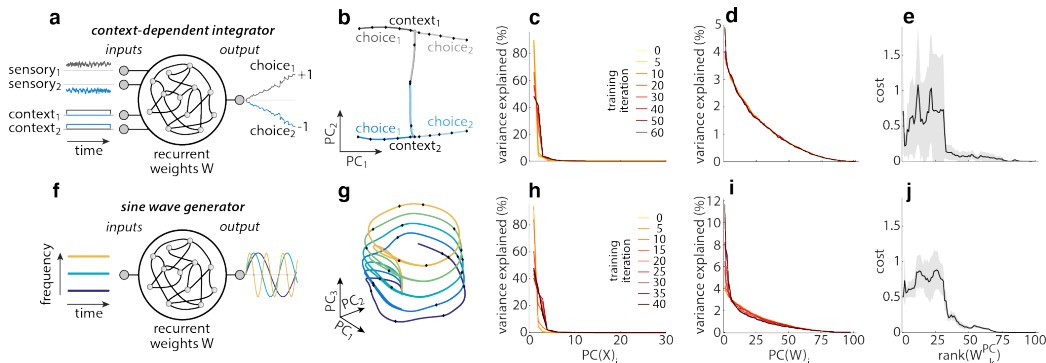

Figure 1: *High-variance dimensions of the connectivity*. (**a**) Task schematic of the context-dependent integrator. (**b**) Low-dimensional projection of condition average trajectories for an example context-dependent integrator. (**c**) Variance explained (in activity space) by individual PCs of the network activity $\mathbf{X}$ over all input conditions, shown at different stages of training. (**d**) Variance explained (in weight space) by individual PCs of the weight matrix $\mathbf{W}$ at different stages of training. (**e**) Network output cost (Eq. 3) of networks with reduced-rank weight matrices $\mathbf{W}_k^{PC}$ for $k = 1 : N$ (Eq. 5), averaged over input conditions (shaded area: median absolute deviation ($mad$) over trials). (**f-j**) Analogous to (**a-e**), but for a sine wave generator. (**b-e**) and (**g-j**) 1 representative network per task.

where $\mathbf{w}_i$ and $\mathbf{v}_i$ are the left and right singular vectors respectively, and $s_i$ the associated singular values. In general, we refer to the i-th left singular vectors of a matrix $\mathbf{M} \in \mathbb{R}^{N \times C}$ ($C \in \mathbb{R}$) as the principal components of the matrix, $PC(\mathbf{M})_i$.

We first ask whether the weight matrices of the trained networks are low or high-dimensional, by assessing the amount of variance in the connectivity ($s_i^2$) explained by principal components $PC(\mathbf{W})_i$. In the trained networks, variance explained falls off slowly over $PC(\mathbf{W})_i$, implying that the weight matrices $\mathbf{W}$ are high-dimensional. The variance explained by individual PCs changes little during training (Fig. 1d, i shown for column dimensions; colors: training iteration) but is strongly influenced by the rank of the untrained network (Fig. S6). Despite having a high-dimensional connectivity, the network activities ($\mathbf{X} = [\mathbf{x}_1, \dots \mathbf{x}_T]$ and $\mathbf{R} = [\mathbf{r}_1, \dots \mathbf{r}_T]$; both further concatenated over all input conditions) are consistently low-dimensional. Activity is low-dimensional even before training, as $\mathbf{W}$ is initialized with a spectral radius of 1 (Fig. 1c, h; colors: training iteration). Throughout training, more than 95% of the variance in the network activities $\mathbf{X}$ can be described by less than ten dimensions. Thus, even though the underlying weight matrix (being high-dimensional) could support activity in any region of state space, the network activity generally only occupies a low-dimensional subspace. This finding raises the question of whether all dimensions in $\mathbf{W}$ are truly necessary to perform the task.

To assess the functional importance of single dimensions of $\mathbf{W}$, we construct reduced-rank approximations of $\mathbf{W}$ from only a subset of all $PC(\mathbf{W})_i$ and then assess the performance of the resulting RNN (as in [22]). Each low-rank approximation is constructed by including only the first $k$ $PC(\mathbf{W})_i$:

$$\mathbf{W}_k^{PC} = \Sigma_{i=1}^k \mathbf{w}_i s_i \mathbf{v}_i^T \tag{5}$$

RNN performance based on $\mathbf{W}_k^{PC}$ is evaluated as above to obtain the network output cost (Eq. 3).

We find that a large number of $PC(\mathbf{W})_i$ are consistently required to reach a similar performance as the original network. For the two example networks, 88 PCs are required to achieve original performance for context-dependent integration, and 86 PCs for sine wave generation (Fig. 1e, j; we define *original performance* as 4 times the cost of the corresponding full-rank RNN, Fig. S7 shows more example networks). Furthermore, performance does not improve monotonically with increasing rank, but rather displays sudden jumps at specific ranks (Fig. 1e, j). These jumps suggest that some high-variance dimensions are more relevant than others in driving the network output. These putative differences in functional relevance are not obviously mirrored in the amount of variance in $\mathbf{W}$ explained by each high-variance dimension (Fig. 1d, i). In other words, the amount of variance in connectivity space explained by a given dimension in $\mathbf{W}$ does not appear to directly correspond to its functional relevance.

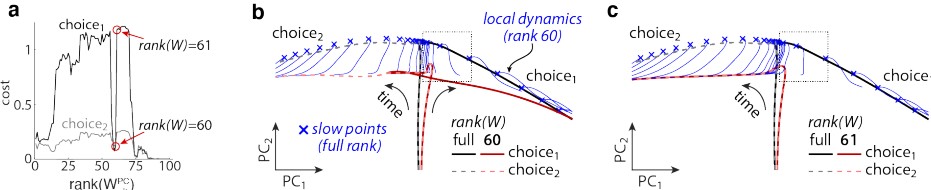

Figure 2: *Local and global dynamics.* (**a**) Network output cost for reduced-rank weight matrices $\mathbf{W}_k^{PC}$ in an example network for context-dependent integration. Analogous to Fig. 1**e**, but shown separately for choice$_1$ and choice$_2$ trials in context$_1$. (**b**) Average trajectories for the conditions in (**a**) in the full-rank and a reduced-rank ($k = 60$) network (grayish and reddish curves, see legend). Blue lines show recurrent dynamics in the reduced-rank network initialized at locations corresponding to slow points of the dynamics in the full-rank network (blue crosses). (**c**) Same as (**b**) but for a reduced-rank network with $k = 61$.

Overall, it appears that the inferred high-variance dimensions may be ill-suited to shed light on the relation between RNN connectivity and dynamics. These findings point to two possible scenarios: the functionally relevant subspace of the weight matrix in these unconstrained RNNs is genuinely high-dimensional, precluding simpler descriptions of the most relevant components of the connectivity; or high-variance dimensions $PC(\mathbf{W})_i$ may in general not be well-aligned with functionally relevant dimensions of the connectivity.

## 2.2 Operative dimensions

### 2.2.1 Definition of operative dimensions

To distinguish between these two scenarios, we devised an approach to directly identify functionally relevant dimensions of the weight matrix. We refer to this new type of dimensions as the *operative dimensions* of the connectivity.

The definition of operative dimensions is based on the key insight that sudden jumps in performance (*cost*, Eq. 3) can be caused by small changes in the local dynamics. Here we illustrate this for one example network trained on context-dependent integration (Fig. 2). We focus on 2 out of 8 input conditions, for which the cost shows prominent jumps at specific ranks $k$, specifically when transitioning from $k = 60$ to $k = 61$ (Fig. 2a; red circles).

To understand this sharp increase, we examine the activity trajectories produced by the original full-rank and the reduced-rank RNN for the two considered input conditions ($k = 60$ in Fig. 2b; $k = 61$ in Fig. 2c). The well-performing, full-rank RNN produces trajectories that start in the center and move out to endpoints on the right (*choice$_1$*; solid black) or left (*choice$_2$*; dashed gray). The RNN with $k = 60$ closely matches these dynamics (Fig. 2b; red curves, solid and dashed). On the other hand, the RNN with $k = 61$ erroneously produces trajectories that move to the left endpoint for both conditions (Fig. 2c; overlapping red curves). This discrepancy between *choice$_1$* trajectories in the original full-rank and $k = 61$ RNN underlies the large jump in cost (Fig. 2a).

Critically, we found that the large difference in cost between the two reduced-rank networks is due to only a small change in the underlying local, recurrent dynamics. To visualize the local dynamics in the two reduced-rank RNNs, we generate brief activity trajectories (Fig. 2b,c; blue curves) initialized at the location of identified slow-points of the dynamics in the original full-rank RNN (blue crosses; identified as in [19]). Here, we isolate the recurrent contribution to the dynamics by running the RNN without any external inputs (Eq. 1 with $\mathbf{u}_t = 0$, $\boldsymbol{\sigma}_t = 0$). The resulting trajectories suggest that the local recurrent dynamics is substantially different across networks only in the region of state space that activity travels through at the onset of the trial (Fig. 2b,c; dashed square). This small local difference is sufficient to redirect the *choice$_1$* trajectories for $k = 61$ towards the wrong endpoint. This mistake cannot be corrected by recurrent dynamics at other locations in state space, resulting in the large cost. In summary, the functional relevance of individual dimensions of $\mathbf{W}$ is hard to assess based on changes in the cost, but rather may be better judged based on its effect on local network dynamics.

Based on this insight, we define *operative dimensions* as dimensions in $\mathbf{W}$ that have a large impact on the local dynamics if removed from $\mathbf{W}$ (Fig. 3a). Given an arbitrary unit vector $\mathbf{a} \in \mathbb{R}^N$, we define $\hat{\mathbf{W}}$ as the matrix of rank $N-1$ obtained by removing the dimension $\mathbf{a}$ from the column space of $\mathbf{W}$ (orthogonal projection):

$$\hat{\mathbf{W}} = \mathbf{W} - \mathbf{a}(\mathbf{a}^T \mathbf{W}) \tag{6}$$

In the absence of noise, the dynamics of the resulting reduced-rank RNN are then given by $\tau \dot{\hat{\mathbf{x}}}_t = -\mathbf{x}_t + \hat{\mathbf{W}} \mathbf{r}_t + \mathbf{B} \mathbf{u}_t$. At location $\mathbf{x}_t$ the activity evolves to $\hat{\mathbf{x}}_{t+1} = \mathbf{x}_t + \dot{\hat{\mathbf{x}}}_t dt$ over one time-step. Likewise, the state of the full-rank network evolves to $\mathbf{x}_{t+1} = \mathbf{x}_t + \dot{\mathbf{x}}_t dt$, where we have set $\boldsymbol{\sigma}_t = 0$ in Eq. 1. We quantify the change in local dynamics brought about by removing $\mathbf{a}$ as:

$$\Delta f = \|\mathbf{x}_{t+1} - \hat{\mathbf{x}}_{t+1}\|_2 \tag{7}$$

as illustrated graphically in Fig. 3a. We then define the operative dimensions of $\mathbf{W}$ based on $\Delta f$ in two steps. In the first step, we infer a set of *local* operative dimensions at each of $P$ sampling locations $\mathbf{y}_j \in \mathbb{R}^N$ in state space. The sampling locations are chosen as an evenly distributed subset of the states $\mathbf{x}_t$ explored by the condition average trajectories of the full-rank network (Fig. 1b, g; only a subset of $P > 100$ locations shown). This choice of sampling locations ensures that we consider only local dynamics that is likely to contribute to solving the task at hand. The first local operative dimension $\mathbf{d}_{1,j}$ at location $\mathbf{y}_j$ is defined as the dimension $\mathbf{a}$ that maximizes $\Delta f$:

$$\mathbf{d}_{1,j} = \underset{\mathbf{a}}{\mathrm{argmax}}(\Delta f)_{\{\mathbf{x}_t = \mathbf{y}_j\}} \tag{8}$$

Up to $N-1$ further local operative dimensions $\mathbf{d}_{i,j}$ at the same location $\mathbf{y}_j$ are defined in the same way, but under the additional constraint that they need to be orthogonal to all previously identified local operative dimensions at that location:

$$\mathbf{d}_{i,j} = \underset{\mathbf{a}}{\mathrm{argmax}}(\Delta f)_{\{\mathbf{x}_t = \mathbf{y}_j\}}, \text{constrained by: } \mathbf{d}_{i,j}^T \mathbf{d}_{i^*,j} = 0, \forall i^* < i \tag{9}$$

In the second step, we define the *global* operative dimensions by combining the local operative dimensions $\mathbf{d}_{i,j}$ from all sampling locations $\mathbf{y}_j$ ($i = 1 : N, j = 1 : P$). Specifically, the local operative dimensions are scaled by their local $\Delta f$ and concatenated into one matrix $\mathbf{L}$.

$$\mathbf{L} = [\mathbf{d}_{1,1} \Delta f_{1,1}, \mathbf{d}_{1,2} \Delta f_{1,2}, \dots, \mathbf{d}_{N,P-1} \Delta f_{N,P-1}, \mathbf{d}_{N,P} \Delta f_{N,P}] \tag{10}$$

where $\Delta f_{i,j} = \Delta f_{\{\mathbf{x}_t = \mathbf{y}_j, \mathbf{a} = \mathbf{d}_{i,j}\}}$. The i-th global operative dimensions $\mathbf{q}_i$ is then defined as the i-th left singular vector of $\mathbf{L}$:

$$\mathbf{L} = \Sigma_{i=1}^N \mathbf{q}_i g_i \mathbf{p}_i^T \tag{11}$$

where $g_i$ are the singular values, and $\mathbf{p}_i$ the right singular vectors of $\mathbf{L}$. The subspace spanned by the global operative dimensions $\mathbf{q}_i$ consists of all left singular vectors with $g_i > 0$. Note that these steps result in operative dimensions of the *column* space of $\mathbf{W}$, which are referred to as *column dimensions* in the figures. We employ an analogous approach to define operative dimensions of the *row* space of $\mathbf{W}$ which yields global operative row dimensions $\bar{\mathbf{q}}_i$ (*row dimensions* in figures; see section A.2.2).

### 2.2.2 Operative dimensions identify a low-d functional subspace of the connectivity

To quantify the functional relevance of the global operative dimensions, we proceed as for the high-variance dimensions above (Eq. 5) by constructing reduced-rank approximations of $\mathbf{W}$ from only a subset of the operative dimensions. For the column dimensions, the reduced-rank approximations $\mathbf{W}_k^{OP}$ are given by:

$$\mathbf{W}_k^{OP} = \Sigma_{i=1}^k \mathbf{q}_i (\mathbf{q}_i^T \mathbf{W}) \tag{12}$$

Network activity $\mathbf{x}_{t,k}^{OP}$ based on $\mathbf{W}_k^{OP}$ is then computed as above (Eq. 1).

Unlike the high-variance dimensions, the global operative dimensions identify a low-dimensional subspace that is sufficient for the RNN to achieve the original performance in both tasks (Fig. 3b-c, e-f; solid lines, qualitatively similar results on sequential MNIST are shown in section A.3.8). We find that 15 column and 27 row dimensions are sufficient to achieve original performance for context-dependent integration; and 29 column and 41 row dimensions for sine wave generation. Thus, the RNNs are *functionally* low-rank even though the underlying weight matrix is high-dimensional.

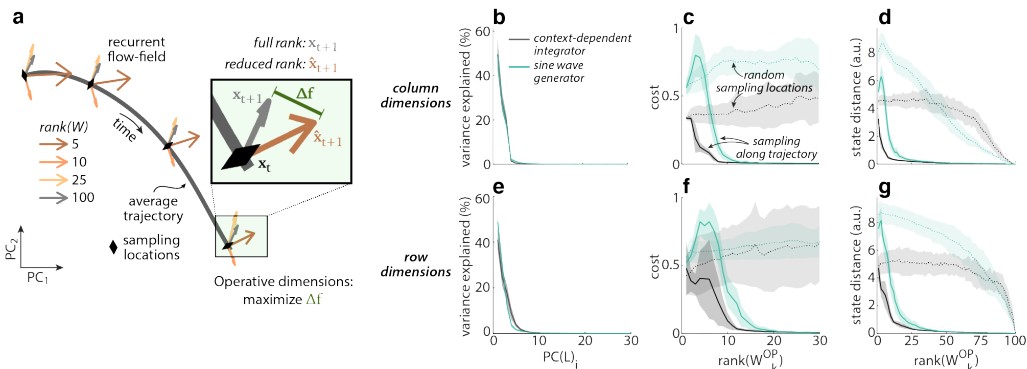

Figure 3: *Operative dimensions.* (**a**) Definition of operative dimensions based on local recurrent dynamics along a condition average trajectory. Arrows show the recurrent contribution to the dynamics for the full-rank and several reduced-rank networks (colors, see legend). Local operative dimensions maximize $\Delta f$. Here inputs $\mathbf{u}_t$ and noise $\boldsymbol{\sigma}_t$ are set to zero. (**b**) Rank of global operative column dimensions, estimated with PC analysis on concatenated local operative dimensions (Eq. 10 and 11). (**c**) Network output cost of networks with reduced-rank weight matrix $\mathbf{W}_k^{OP}$ for $k = 1 : N$ (Eq. 12). (**d**) State distance between trajectories in the full-rank network and in networks with reduced-rank weight matrix $\mathbf{W}_k^{OP}$. (**b-d**) Based on global operative *column* dimensions and averaged over 20 networks per task; shaded area: $mad$. Network output cost obtained with internal and input noise, state distance without any noise. (**e-g**) Same as (**b-d**) for global operative *row* dimensions.

This clear difference between the global operative dimensions and the high-variance dimensions is also reflected in their weak alignment to each other (Fig. S8). This finding implies that much of the network connectivity in these trained networks is not required to perform the task.

Notably, the identified functionally relevant subspace of the connectivity is sufficient to generate the original network trajectory, not just the network activities along the output direction. The average distance between the network trajectories of the reduced-rank and the full-rank networks decreases rapidly as the global operative dimensions are sequentially added to the weight matrix (Fig. 3d, g; $state\ distance = \frac{1}{T} \sum_{t=1}^{T} \|\mathbf{x}_t - \mathbf{x}_{t,k}^{OP}\|_2$). This observation follows from the definition of operative dimensions, which focuses not on changes in network output, but rather on the recurrent dynamics at sampling locations lying along the entirety of the condition average trajectories. We obtained similar results for networks at different stages of training (Fig. S14) and with different types of connectivity and architecture (Fig. S15). Similarly, even though operative dimensions are defined based on how much they alter the local dynamics when removed from $\mathbf{W}$, the first few operative dimensions are sufficient to preserve the dominant local, linear dynamics (Fig. S17).

Unlike for high-variance dimensions (Fig. 1e, j), RNN performance and state distance change smoothly with increasing rank of the global operative dimensions (Fig. 3c-d, f-g). However, adding the first few dimensions to the weight matrix often hurts performance. This effect can happen because the global operative dimensions are not sorted based on when they are required during individual trials, but rather by their impact on local network dynamics. Indeed, the first few operative dimensions are mostly required at state-space locations explored late in the trial, but are insufficient to sustain the dynamics required early in the trial (Fig. S21c, f, i, l).

Identifying the operative dimensions critically relies on the correct choice of sampling locations. First, we illustrate the importance of the choice of sampling locations by defining global operative dimensions based on random sampling locations in state space (Fig. 3c, f; dotted lines). For the same choice of rank, the resulting dimensions yield much poorer performance compared to the operative dimensions defined as above, emphasizing the importance of capturing the local network dynamics within the functionally relevant part of state space. Second, even when sampling locations are defined along the condition average trajectories, they need to be sampled at high enough density. When too few sampling locations are chosen along the trajectories, the identified global operative dimensions are less effective at reproducing network output and trajectories (Fig. S18).

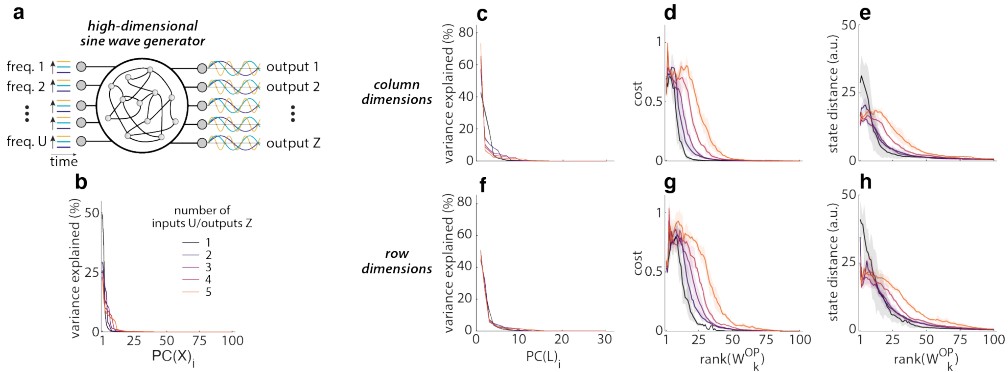

Figure 4: *Operative dimensions and network complexity.* (**a**) Task schematic for high-dimensional sine wave generator networks trained to output 1-5 sine waves simultaneously. (**b**) Variance explained (in activity space) by individual PCs of the network activity $\mathbf{X}$ over all input conditions. (**c**) Rank of global operative column dimensions, estimated with PC analysis on concatenated local operative dimensions $\mathbf{L}$ (Eq. 10 and 11). (**d**) Network output cost of networks with reduced-rank weight matrix $\mathbf{W}_k^{OP}$ for $k = 1 : N$ (Eq. 12). (**e**) State distance between trajectories in the full-rank network and in networks with reduced-rank weight matrix $\mathbf{W}_k^{OP}$. (**c-e**) Based on global operative *column* dimensions; averaged over 5 networks per number of inputs/outputs; shaded area: $mad$. Network output cost obtained with internal and input noise, state distance without any noise. (**f-h**) Same as (**c-e**) for global operative *row* dimensions.

One consequence of the tight link between local network dynamics and operative dimensions is that the number of operative dimensions required to approximate a particular network may increase with the complexity of the network activity, and of its inputs and outputs [23]. To illustrate this relation, we systematically varied the complexity of RNN computations by training sine wave generators with varying numbers of input and output signals (between $1 - 5$ inputs $U$ and outputs $Z$; Fig. 4). As expected, the network activity $\mathbf{x}_t$ becomes increasingly high-dimensional for increasing values of $U$ and $Z$ (Fig. 4b), and correspondingly the number of global operative dimensions required to achieve the original performance increases as well (Fig. 4c-e and f-h). We obtained a similar effect when increasing the dimensionality of the inputs into the RNNs, while keeping the dimensionality of the output fixed. (Fig. 13). Analytical considerations also imply that, in vanilla RNNs described by Eq. 1, the number of global operative dimensions is tightly linked to the dimensionality of the network activity $\mathbf{R}$ (section A.2.4 and A.2.5).

### 2.2.3 Operative dimensions relate functional modules to weight subspaces

Past studies have shown that RNNs can implement complex, context-dependent computations by "tiling" activity state space into separate functional modules [21, 19, 24]. Dynamics within individual modules is often approximately linear, but different approximately linear dynamics, corresponding to different input-output relations, are implemented across modules [20].

Our definition of operative dimensions is well-suited to ask whether the existence of functional modules in activity space has a correspondence at the level of the structure of the connectivity. To obtain a more fine-grained mapping from function to structure, the global operative dimensions can be generated based on different subsets of local operative dimensions. Specifically, sampling locations can be grouped based on their functional meaning, i.e. based on which condition average they belong to. The resulting sets of function-specific global operative dimensions can then link different network functions to particular subspaces in the weight matrix.

We demonstrate this approach for the context-dependent integrator network. We inferred global operative dimensions from local operative dimensions that were collected either separately by context, but pooled over choice, or separately by choice, but pooled over context (details section A.2.6). We refer to the resulting context-dependent global operative column dimensions as $\mathbf{q}_i(context_j)$, and the choice-dependent dimensions as $\mathbf{q}_i(choice_j)$ ($i = 1 : N$, $j = 1 : 2$; row dimensions: $\bar{\mathbf{q}}_i$). We compared these dimensions directly by measuring their pairwise alignment (subspace angle,

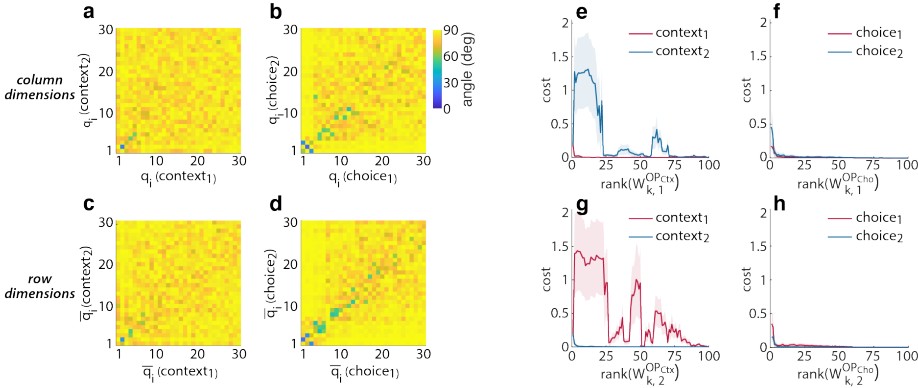

Figure 5: *Function-specific global operative dimensions.* (**a**) Subspace angle between global operative column dimensions $\mathbf{q}_i$ defined separately for context$_1$ and context$_2$. (**b**) Subspace angle between global operative column dimensions $\mathbf{q}_i$ defined separately for choice$_1$ and choice$_2$. (**c-d**) Same as (**a-b**) for global operative *row* dimensions $\overline{\mathbf{q}}_i$. (**e**) Network output cost averaged separately over trials of context$_1$ or context$_2$ for networks with reduced-rank weight matrix consisting of the first $k$ function-specific global operative column dimensions from context$_1$ ($\mathbf{W}_{k,j}^{OP_{Ctx}}$, context $j = 1$). (**f**) Network output cost averaged separately over trials of choice$_1$ or choice$_2$ for networks with reduced-rank weight matrix consisting of the first $k$ function-specific global operative column dimensions from choice$_1$ ($\mathbf{W}_{k,j}^{OP_{Cho}}$, choice $j = 1$). (**g**) Same as (**e**), but based on global operative column dimensions from context 2. (**h**) Same as (**f**), but based on global operative column dimensions from choice 2.

Fig. 5a-d). We find that the function-specific global operative dimensions are only weakly aligned across the two contexts, but more strongly aligned across choices, suggesting that the RNN uses different weight subspaces to implement the sensory integration in each context, but reuses weight structures to implement the sensory integration across choices.

To further support this conclusion, we checked if the global operative dimensions inferred from one functional module can support accurate computations in another module. We first constructed module specific, reduced-rank approximations of $\mathbf{W}$ as above (Eq. 12), which we refer to as $\mathbf{W}_{k,j}^{OP_{Ctx}}$ (context-specific) and $\mathbf{W}_{k,j}^{OP_{Cho}}$ (choice-specific). We then measured network performance across conditions when using these different reduced-rank approximations (Fig. 5e-h). In agreement with the subspace-angles above, we find that networks based on operative dimensions defined in context$_1$ perform poorly in context$_2$, and vice versa (Fig. 5e, g) whereas operative dimensions from a given choice yield comparatively good performance also in the other choice (Fig. 5f, h). These observations are in line with previous findings [19] that the context-dependent integrator implements an approximate line attractor in each context, with mostly preserved dynamics along a given line attractor (i.e. across choices), but rather different dynamics between attractors (i.e. across context). Interestingly, the identified structure in the underlying connectivity, while clearly revealed by the operative dimensions (Fig. 5), does not become apparent when studying the weight matrix or contributions of individual network units directly (Fig. S19).

## 3 Discussion

In this work, we present an approach for inferring the *operative dimensions* of an arbitrary weight matrix in an RNN. The operative dimensions of the connectivity are defined based on their impact on the computationally relevant, local recurrent dynamics of the network. We find that for the examined tasks, the operative dimensions span a low-dimensional subspace of the connectivity space that is sufficient to produce accurate outputs in a reduced-rank network.

**Relation to state of the art.** Our framework of operative dimensions extends recent work based on RNNs that by construction are explicitly low-rank, which already showed that many tasks of relevance to neuroscience can be solved with RNNs that rely entirely on low-rank connectivity [17].

Unlike in this previous work, the networks we analyzed have a connectivity that is not constrained–the dimensionality of the underlying weight matrix is high at initialization, and remains high throughout training (Fig. 1d, i). This high dimensionality persists even though the weight updates brought about by training are low-rank [22]. However, our analyses showed that only a low-dimensional subspace of the effectively high-dimensional connectivity is used to solve the task, whereby the remainder of the structure in the weight matrix plays essentially no role in solving the task at hand. This finding opens the possibility to apply the conceptual framework of low-rank networks [17], which has provided valuable and direct insights into the relation of connectivity and dynamics, also to other types of RNNs.

Operative dimensions amount to a form of model reduction [25, 26] that emphasizes the preservation of attractor dynamics [27, 28] for the special case of a model parameterized as a neural network. Critically, our approach to relating the model structure to its function relies on characterizing the dynamics locally, along explored activity trajectories, similarly to methods to quantify non-linear dynamics based on Lyapunov or bred vectors [29, 30, 31].

More generally, our work makes a contribution towards increasing the *interpretability* of artificial neural networks. Past work had shown that unconstrained, trained RNNs need not be considered as a "black box", but rather that the computations implemented by many RNNs can be understood at the level of population dynamics, through the interaction of inputs and dynamical primitives like attractors and saddle points [20, 32, 24]. Recent work related the implemented dynamical primitives to the underlying connectivity [17] in constrained RNNs. We have shown that establishing such relations is in principle also possible through the definition of condition-dependent operative dimensions (Fig. 5) without any specific constraints on the connectivity. Current opinions assume that such increased understanding and interpretability of artificial networks is desirable both to increase acceptance of the resulting machine learning applications throughout society, but also potentially to design better artificial system that overcome biases and limitations resulting from current approaches (see [33, 34, 35] for reviews). Nonetheless, potentially negative societal impacts of increased interpretability cannot be ruled out. For instance, if used on RNNs trained on personal data, operative dimensions may potentially facilitate the extraction of private information that was otherwise hidden in high-dimensional weight matrices.

**Limitations.** One limitation of our approach to identifying operative dimensions is that it relies on studying local recurrent dynamics based on a somewhat arbitrary definition of sampling locations in activity state space. By design, the inferred operative dimensions will be sufficient to reproduce the full-rank dynamics only at these specific locations in state space. Note that there is no explicit requirement for the operative dimensions to reproduce the desired network output. We picked sampling locations along the condition average trajectories, based on the assumption that these average trajectories provide a good coverage of all the relevant local dynamics. In practice, this assumption may not hold in all RNNs. For one, average trajectories may travel through state-space regions that are not visited by any single trials, leading one to under-sample functionally relevant regions. For another, single-trial and average trajectories may also reflect dynamics that is not functionally relevant, and thus lead to over-sampling of regions that are not directly involved in generating the output. Further work may address alternative approaches for choosing sampling locations to optimize their functional relevance. To some extent, the adequacy of sampling locations can be tested empirically by systematically varying the number and location of sampling locations to optimize the performance of the inferred reduced-rank networks (Fig. S18).

Another potential limitation of our approach is that it may not be equally effective in identifying a functional subspace of the connectivity across all types of RNNs and in more complex tasks. While here we have focused on vanilla RNNs, we found that a low-dimensional functional subspace can be identified in such RNNs for a variety of network architectures, including with non-overlapping populations of excitatory and inhibitory neurons, or the use of different non-linear activation functions (Fig. S15). The properties of learned dynamics in the kind of tasks we employed is largely conserved across different types of networks (GRU, LSTM; [21, 24]), in particular the role of "tiling" activity space into distinct computational modules. This observation implies that our approach to relating connectivity and function at the level of local dynamics is at least meaningfully applicable even in these different kinds of networks. While we find that operative dimensions can retrieve a functionally relevant subspace also for a richer task like sequential MNIST (Fig. S16), it remains an open question whether these approaches will extend to harder AI problems.

**Conclusion** Operative dimensions can infer a functionally relevant subspace within high-dimensional RNN connectivity that is sufficient to perform the task at hand. On one hand, this observation might benefit practical applications of RNNs as a computational tool, e.g. for continual learning in RNNs, by specifically protecting functionally relevant subspaces of the weight matrix to avoid catastrophic forgetting [36], or for weight compression, by storing the weight matrix as a linear combination of the global operative dimensions [37, 22]. On the other hand, the ability to identify functionally specific subspaces in the network connectivity may improve the applicability of RNNs as a model in neuroscience, as it simplifies the critical challenge of linking the properties of the connectivity to the network dynamics and may thus provide guidance on how to relate structure to function in complex biological datasets.

## Code Availability

We used Matlab to perform the data analysis. All the code and models required to reproduce the main analyses supporting our conclusions are available in matlab and python at: `https://gitlab.com/neuroinf/operativeDimensions`

## Author Contributions

R.K. and V.M. conceived the idea for the study, R.K. performed the analyses. M.C. and G.I. provided feedback throughout the project. R.K. and V.M. wrote the manuscript. R.K., V.M., M.C. and G.I. reviewed the manuscript.

## Acknowledgements

We thank all group members (Neural Computation and Cognition (V.M.), Cortical Computation (M.C.) and Neuromorphic Cognitive Systems (G.I.)) for their valuable feedback and discussions.

## Funding

This work was supported by the EU-H2020 FET project CResPACE (Grant No. 732170, G.I.), grants from the Simons Foundation (SCGB 328189, V.M.; SCGB 543013, V.M.) and the Swiss National Science Foundation (SNSFPP00P3_157539, V.M.).

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
