# A  Appendix

## A.1  Recurrent neural networks

### A.1.1  Task structures

**Context-dependent integration**   The context-dependent integrator [19] receives four inputs: two time-varying, noisy sensory inputs of six different levels, drawn at each time from $\mathcal{N}(\mu, \sigma = 31.623\sqrt{dt} \approx 1)$ with $\mu \in \{-0.5, -0.12, -0.3, 0.03, 0.12, 0.5\}$; and two constant context inputs, set to 1 or 0. In context$_1$, input context$_1$ is set to 1 and input context$_2$ is set to 0 while both sensory inputs are ON. The task is to integrate the relevant sensory input (input sensory$_1$) over time in the activity of the output unit $\mathbf{z}_t$ and ignore the irrelevant sensory input (input sensory$_2$) and vice versa for context$_2$. Each trial consists of $650\,ms$ burn period (only input context$_j$=1; both sensory inputs OFF) followed by $750\,ms$ sensory integration time (only input context$_j$=1; both sensory inputs ON). One trial has $T = 1400$ time steps.

**Sine wave generation**   The sine wave generator [20] receives one constant input which can be set to one of 51 different levels $u_t = l/51 + 0.25\ \forall t < T, l \in \mathbb{R}$. The level of the constant input defines the target frequency of the sine wave generated in the activity of the network output unit $\mathbf{z}_t$ whereby the target frequencies $\omega_l$ $(l = 1:51)$ are equally spaced between $1 - 6\,rad/sec$ to define $\omega_l = 0.1 * (l - 1) + 1$. Each trial lasts $500\,ms$ during which the input $u_t$ is constantly ON ($T = 500$).

**High-dimensional sine wave generation**   The high-dimensional sine wave generator network (Fig. 4) is an extension of the sine wave generator network explained in section A.1.1 [20]. It receives $U = 1:5$ constant inputs $\mathbf{u}_t$ and is trained to generate sine wave activity in $U = Z = 1:5$ outputs $\mathbf{z}_t$. The $U$ constant inputs are set independently of each other to one of 51 different levels $u_t(i) = l_i/51 + 0.25\ \forall t < T, i = 1:U, l_i \in \mathbb{R}$. The level of the i-th constant input $u_t(i)$ defines the target frequency of the sine wave generated in the activity of the i-th network output unit $z_t(i)$ whereby the target frequencies $\omega_{l_i}$ $(l_i = 1:51)$ are equally spaced between $1 - 6\,rad/sec$ to define $\omega_{l_i} = 0.1 * (l_i - 1) + 1$. Each trial lasts $500\,ms$ during which the input is constantly ON ($T = 500$). For $U = Z = 1$ the high-dimensional sine wave generation network is identical to the sine wave generation network explained in section A.1.1. We train separate networks for every number of $U = Z = 1:5$.

### A.1.2  Network training

In both tasks, all weights ($\mathbf{B}, \mathbf{W}, \mathbf{Y}$) are optimized using Hessian-free optimization [38] to minimize the network output cost:

$$cost_{training} = \frac{1}{Z}\Sigma_{i=1}^{Z}\Sigma_{t=1}^{T}(z_t^*(i) - z_t(i))^2 \tag{13}$$

between the respective target activity $\mathbf{z}_t^*$ and the network output activity $\mathbf{z}_t$ at every time point during the trial, averaged over all input conditions (figures show $cost_{training}$ scaled by $1/T$, Eq. 3). $\mathbf{B}, \mathbf{W}$, and $\mathbf{Y}$ are randomly initialized and set to have a spectral radius of 1 ($\mathbf{B}_0 = \mathcal{N}(0, 1)$, $\mathbf{W}_0 = \mathcal{N}(0, 1/\sqrt{N})$, $\mathbf{Y}_0 = \mathcal{N}(0, 1)$). All input conditions are trained simultaneously with equal probability to ensure a balanced training set (batch size = 400). The presented results are obtained using the same input conditions as during training but with varying input noise and internal noise. The code to train and run the RNNs was modified from [19]. For the main results we trained 20 networks on context-dependent integration, 20 networks on sine wave generation and 25 networks on high-dimensional sine wave generation (5 networks per $U = Z$ ($U = 1:5$, $Z = 1:5$)).

## A.2  Definition of operative dimensions

### A.2.1  Sampling locations

We place the sampling locations $\mathbf{y}_j$ $(j = 1:P)$ equally spaced in time on the condition average trajectories of each network. The condition average trajectories are defined as the mean network activity per time point of the corresponding input condition, averaged over input and internal noise (20 trials per condition).

For the context-dependent integration network we considered the condition average trajectories for 8 different input conditions whereby we define the input condition based on the context (1 or 2), the choice (1 or 2) and the coherency of the sensory inputs (congruent (sign(input sensory$_1$) = sign(input sensory$_2$)) or incongruent (sign(input sensory$_1$) != sign(input sensory$_2$)) ) in every trial ($2^3 = 8$ distinct input conditions). We placed the sampling locations equally spaced in time along each of these 8 condition average trajectories at every 100-th time step (t=1:100:1400 resulting in 15 sampling locations per condition average trajectory) to obtain $\mathbf{y}_1 = \mathbf{x}_1$, $\mathbf{y}_2 = \mathbf{x}_{100}$, $\mathbf{y}_3 = \mathbf{x}_{200}$, ...for every input condition. In total, we defined $P = 8 \times 15 = 120$ sampling locations.

The sine wave generation network has 51 different input conditions (51 input levels). For the definition of local operative dimensions, we subsampled the input conditions and only considered every 5-th input level ($l = 1 : 5 : 51$), resulting in 11 condition average trajectories. We placed the sampling locations equally spaced in time along each of these 11 condition average trajectories at every 50-th time step (t=1:50:500 resulting in 11 sampling locations per condition average trajectory) to obtain $\mathbf{y}_1 = \mathbf{x}_1$, $\mathbf{y}_2 = \mathbf{x}_{50}$, $\mathbf{y}_3 = \mathbf{x}_{100}$, ...for every input condition. In total, we defined $11 \times 11 = 121$ sampling locations.

### A.2.2 Operative row dimensions

Analogously to the operative *column* dimensions, the operative *row* dimensions are defined as the dimensions in $\mathbf{W}$ that have a large impact on the local dynamics if removed from the *row* space of $\mathbf{W}$ (Fig. 3a).

Given an arbitrary unit vector $\mathbf{a} \in \mathbb{R}^N$, $\|\mathbf{a}\|_2 = 1$, we define $\hat{\mathbf{W}}$ for the operative *row* dimensions as the matrix of rank $N - 1$ obtained by removing the dimension $\mathbf{a}$ from the *row* space of $\mathbf{W}$:

$$\hat{\mathbf{W}} = \mathbf{W} - (\mathbf{a}(\mathbf{W}\mathbf{a})^T)^T \tag{14}$$

The respective local and global operative dimensions are defined as explained in section 2.2.1. Based on Eq. 7, 8, 9, 10, 11, the i-th global operative row dimension is defined as the i-th left singular vector of $\mathbf{L}$ and we refer to is as $\bar{\mathbf{q}}_i$.

The reduced-rank approximation of $\mathbf{W}$ constructed from only a subset of the global operative *row* dimensions is then given by:

$$\mathbf{W}_k^{OP} = \Sigma_{i=1}^k (\bar{\mathbf{q}}_i(\mathbf{W}\bar{\mathbf{q}}_i)^T)^T \tag{15}$$

For simplicity and readability purposes, below we use the variables for the global operative *column* dimensions throughout the text. All corresponding statements similarly apply also to the row dimensions, unless explicitly stated otherwise.

### A.2.3 Properties of local operative dimensions in vanilla RNNs

For the special case of a vanilla RNN (Eq. 1), several properties of the local and global operative dimensions can be derived analytically. While the derivations below do not apply to other RNN architectures (e.g. LSTM [39], GRU [40]) the general approach and definitions for the estimation of operative dimensions are applicable irrespective of the RNN architecture.

**Analytical derivation of local operative column dimensions** The first local operative dimension $\mathbf{d}_{1,j}$ at location $\mathbf{y}_j$ is the solution of the following optimization problem (Eq. 8):

$$\mathbf{d}_{1,j} = \underset{\mathbf{a}}{\mathrm{argmax}}(\Delta f)_{\{\mathbf{x}_t = \mathbf{y}_j\}}$$

with $\mathbf{a} \in \mathbb{R}^N$, $\|\mathbf{a}\|_2 = 1$, and $\Delta f$ as (Eq. 1, 7):

$$\Delta f = \|\mathbf{x}_{t+1} - \hat{\mathbf{x}}_{t+1}\|_2$$

For a vanilla RNN:

$$\mathbf{x}_{t+1} = \mathbf{x}_t - \mathbf{x}_t \frac{dt}{\tau} + \mathbf{W}\mathbf{r}_t \frac{dt}{\tau} + \mathbf{B}\mathbf{u}_t \frac{dt}{\tau}$$

$$\hat{\mathbf{x}}_{t+1} = \mathbf{x}_t - \mathbf{x}_t \frac{dt}{\tau} + \hat{\mathbf{W}}\mathbf{r}_t \frac{dt}{\tau} + \mathbf{B}\mathbf{u}_t \frac{dt}{\tau}$$

To derive the first local operative *column* dimension this can be simplified as follows:

$$\mathbf{d}_{1,j} = \underset{\mathbf{a}}{\mathrm{argmax}}(\|\mathbf{x}_{t+1} - \hat{\mathbf{x}}_{t+1}\|_2)$$

$$\mathbf{d}_{1,j} = \underset{\mathbf{a}}{\mathrm{argmax}}(\|[\mathbf{x}_t - \mathbf{x}_t\frac{dt}{\tau} + \mathbf{W}\mathbf{r}_t\frac{dt}{\tau} + \mathbf{B}\mathbf{u}_t\frac{dt}{\tau}] - [\mathbf{x}_t - \mathbf{x}_t\frac{dt}{\tau} + \hat{\mathbf{W}}\mathbf{r}_t\frac{dt}{\tau} + \mathbf{B}\mathbf{u}_t\frac{dt}{\tau}]\|_2)$$

$$\mathbf{d}_{1,j} = \underset{\mathbf{a}}{\mathrm{argmax}}(\|\frac{dt}{\tau}(\mathbf{W}\mathbf{r}_t - \hat{\mathbf{W}}\mathbf{r}_t)\|_2)$$

We replace $\hat{\mathbf{W}}$ using Eq. 6:

$$\mathbf{d}_{1,j} = \underset{\mathbf{a}}{\mathrm{argmax}}(\|\frac{dt}{\tau}(\mathbf{W}\mathbf{r}_t - (\mathbf{W} - \mathbf{a}(\mathbf{a}^T\mathbf{W}))\mathbf{r}_t)\|_2)$$

$$\mathbf{d}_{1,j} = \underset{\mathbf{a}}{\mathrm{argmax}}(\|\frac{dt}{\tau}(\mathbf{a}\mathbf{a}^T\mathbf{W}\mathbf{r}_t)\|_2)$$

which has a unique solution for a vector $\mathbf{a}$ that is aligned to $\mathbf{W}\mathbf{r}_t$:

$$\mathbf{d}_{1,j} = \mathbf{W}\mathbf{r}_t \tag{16}$$

**Dimensionality of local operative column dimensions**  From the above derivation also follows that, at any given location $\mathbf{x}_t$ in the state space of a vanilla RNN, only a single local operative column dimension can be derived. This follows from the observation that removal from $\mathbf{W}$ of any vector that is orthogonal to $\mathbf{d}_{1,j} = \mathbf{W}\mathbf{r}_t$ necessarily results in $\Delta f = 0$, and thus does not cause any change in the network dynamics:

$$\Delta f = \|\mathbf{x}_{t+1} - \hat{\mathbf{x}}_{t+1}\|_2$$

$$\Delta f = \|\frac{dt}{\tau}(\mathbf{a}\mathbf{a}^T\mathbf{W}\mathbf{r}_t)\|_2$$

Replacing $\mathbf{W}\mathbf{r}_t = \mathbf{d}_{1,j}$:

$$\Delta f = \|\frac{dt}{\tau}(\mathbf{a}\mathbf{a}^T\mathbf{d}_{1,j})\|_2$$

Given that all local operative column dimensions $\mathbf{d}_{i,j} \ \forall i > 1$ have to fulfill $\mathbf{d}_{i,j}^T\mathbf{d}_{1,j} = 0$ (see Eq. 9), it follows that $\Delta f_{i,j} = 0, \ \forall i > 1$

Hence, when using the standard RNN equation as described in Eq. 1, only a single local operative *column* dimensions $\mathbf{d}_{1,j} = \mathbf{W}\mathbf{r}_t$ can be inferred at any given location in state space.

The previous two key derivations on local operative column dimensions can further be extended to the case where the non-linear RNN dynamics is well described by the local linear approximation $\mathbf{A} \in \mathbb{R}^{N \times N}$ around slow points of the full-rank RNN $\mathbf{x}^* \in \mathbb{R}^N$. Hence, the derivations and corresponding results are more general and apply (approximately) to any RNN architecture that results in dynamics that are locally linear, not just to vanilla RNN.

### A.2.4   Dimensionality of global operative *column* dimensions

The above properties of the *local* operative column dimensions have implications for the overall dimensionality of the *global* operative column dimensions. Specifically, in a vanilla RNN the dimensionality of the subspace spanned by the global operative column dimensions is bounded by the dimensionality of the network activity $\mathbf{r}_t$. This follows from the above derivation (section A.2.3) that there is only a single local operative column dimension at any sampling location $\mathbf{y}_j$ in state space, namely $\mathbf{d}_{1,j} = \mathbf{W}\mathbf{r}_t$ with $\mathbf{r}_t = tanh(\mathbf{y}_j)$.

The population activity $\mathbf{r}_t$ at a given location in state space and time can be written as a linear combination of the principal components of the population activity $PC(\mathbf{R})_i$:

$$\mathbf{r}_t = c_{1,t}PC(\mathbf{R})_1 + c_{2,t}PC(\mathbf{R})_2 + ... + c_{N,t}PC(\mathbf{R})_N$$

where the coefficients $c_{i,t}$ ($i = 1 : N$) depend on state space location and time $t$ in the trial. Combining the above expression with $\mathbf{d}_{1,j} = \mathbf{W}\mathbf{r}_t$ results in:

$$\mathbf{d}_{1,j} = \mathbf{W}(c_{1,t}PC(\mathbf{R})_1 + c_{2,t}PC(\mathbf{R})_2 + ... + c_{N,t}PC(\mathbf{R})_N)$$

$$\mathbf{d}_{1,j} = c_{1,t}(\mathbf{W}PC(\mathbf{R})_1) + c_{2,t}(\mathbf{W}PC(\mathbf{R})_2) + ... + c_{N,t}(\mathbf{W}PC(\mathbf{R})_N)$$

While the coefficients $c_{i,t}$ vary over sampling locations $\mathbf{r}_t = tanh(\mathbf{y}_j)$, the vectors $\mathbf{W}PC(\mathbf{R})_1$, $\mathbf{W}PC(\mathbf{R})_2$, ... ,$\mathbf{W}PC(\mathbf{R})_N$ do not, and rather are a fixed property of an RNN with weight matrix $\mathbf{W}$. Hence, all local operative column dimensions are a linear combination of the $\mathbf{W}PC(\mathbf{R})_i$ with $c_{i,t} > 0$. As a consequence, if the population activity $\mathbf{R}$ is contained in a low-dimensional subspace, the subspace of the column space of $\mathbf{W}$ that is required to perform the task (spanned by the operative column dimensions) is also low-dimensional, and its dimensionality is at most as high as the dimensionality of the responses $\mathbf{R}$, independently of the training procedure. Note that while the dimensionality of these two subspaces is related, the two subspaces need not be overlapping.

Notably, a comprehensive understanding of the exact factors determining the dimensionality of activity in trained RNNs is currently lacking. The dimensionality of the inputs can be expected to be an important factor [1] although in general not the only one. Indeed, RNNs that receive high-dimensional inputs can nonetheless generate low-dimensional dynamics [32]. On the other hand, reservoir computing networks can generate high-dimensional dynamics even when driven with low-dimensional inputs [41, 42]. Our result that the dimensionality of activity in the N-fold sine-wave generator increases with N (Fig. 4b) could further be interpreted as being driven by the dimensionality of the output.

### A.2.5 Dimensionality of global operative *row* dimensions

Analytical constraints to the functional subspace spanned by the global operative *row* dimensions can also be derived. Specifically, this functional subspace is contained within the intersection of the subspace spanned by the network activity $\mathbf{R}$ and the subspace spanned by the row dimensions of $\mathbf{W}$. Indeed, any vector orthogonal to this intersection can be removed from $\mathbf{W}$ without any effect on the network dynamics.

The effect of removing dimension $\mathbf{a}$ from the row space of $\mathbf{W}$ is given by:

$$\overline{\Delta f} = \|\frac{dt}{\tau}(\mathbf{x}_{t+1} - \bar{\hat{\mathbf{x}}}_{t+1})\|_2$$

$$\overline{\Delta f} = \| - \frac{dt}{\tau}((\mathbf{a}(\mathbf{W}\mathbf{a})^T)^T\mathbf{r}_t)\|_2$$

$$\overline{\Delta f} = \| - \frac{dt}{\tau}(\mathbf{W}\mathbf{a}\mathbf{a}^T\mathbf{r}_t)\|_2$$

The term $\mathbf{W}\mathbf{a}$ vanishes for any $\mathbf{a}$ outside the row space of $\mathbf{W}$. The term $\mathbf{a}^T\mathbf{r}_t$ vanishes for any term orthogonal to $\mathbf{r}_t$. Hence, $\overline{\Delta f} = 0$ for any $\mathbf{a}$ outside the intersection between the row space of $\mathbf{W}$ and the activity subspace.

### A.2.6 Function-specific global operative dimensions

To define function-specific global operative dimensions, we combine local operative dimensions only from specific subsets of sampling locations. In Fig. 5, we created four different types of such function-specific global operative dimensions $\mathbf{q}_i(function)$ ($i = 1 : N$) for the context-dependent integration network:

- $\mathbf{q}_i(context_1)$: sampling locations $y_j$ along the condition average trajectories of $context_1$
- $\mathbf{q}_i(context_2)$: sampling locations $y_j$ along the condition average trajectories of $context_2$
- $\mathbf{q}_i(choice_1)$: sampling locations $y_j$ along the condition average trajectories of $choice_1$
- $\mathbf{q}_i(choice_2)$: sampling locations $y_j$ along the condition average trajectories of $choice_2$

All four types combine sampling locations from 4 condition average trajectories with 15 sampling locations each ($P = 60$).

### A.2.7 Required computational resources

Computing the local operative column dimensions is computationally inexpensive because there is an explicit solution ($\mathbf{d}_{1,j} = \mathbf{W}\mathbf{r}_t$; see section A.2.3) and hence it is not required to run the numerical optimization procedure but only 1 matrix multiplication per sampling location.

To obtain the local operative row dimensions we perform a numerical optimization (matlab function *fminunc* to *find the minimum of an unconstrained multivariable function* using the *quasi-newton* optimization algorithm; as provided in the code to obtain the operative dimensions). We run the optimization upto $N$-times per sampling location or until $\Delta f < 1e^{-8}$. It takes less than one minutes to obtain the local operative row dimensions at a given sampling location on a standard machine (6 core, Intel Core i7-7800X CPU, 3.50GHz).

## A.3 Additional analyses and figures

### A.3.1 Importance of initial rank of W

The trained weight matrices $\mathbf{W}$ are generally high-dimensional in both tasks (Fig. 1d, i). Interestingly, the rank of the trained weight matrix seems highly dependent on the rank of the initial weight matrix $\mathbf{W}_0$ (Fig. S6) whereby a low-dimensional, initial weight matrix $\mathbf{W}_0$ generally results in a low-dimensional, trained weight matrix $\mathbf{W}$.

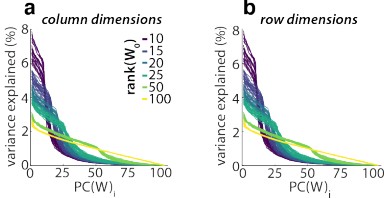

Figure 6: *Dimensionality of trained weight matrices with different initial ranks.* (**a**) Variance explained (in weight space) by individual PCs of the weight matrix $\mathbf{W}$ after training when the initial weight matrix $\mathbf{W}_0$ was randomly initialized with different ranks (colors, see legend). (**b**) Same as (**a**) for row dimensions of $\mathbf{W}$. (**a** - **b**) Each line is one network; 20 networks per $\mathbf{W}_0$; trained on context-dependent integration. For the column as well as the row space, the rank of $\mathbf{W}$ changes little over training, and is instead mainly determined by the rank of $\mathbf{W}_0$. The dimensionality of the weight matrix $\mathbf{W}$ after training is thus only weakly related to the trained task in these cases.

### A.3.2 High-variance dimensions

To assess the functional importance of the high-variance dimensions of $\mathbf{W}$, we sequentially remove the high-variance dimensions from $\mathbf{W}$ while measuring the performance of the reduced-rank network (Fig. 1e, j). Generally, the network performance shows sudden jumps at specific ranks which are hard to interpret. Here we show how the network performance varies over reduced-rank $\mathbf{W}_k^{PC}$ for more example networks to illustrate the large differences that are seen across individual networks (Fig. S7).

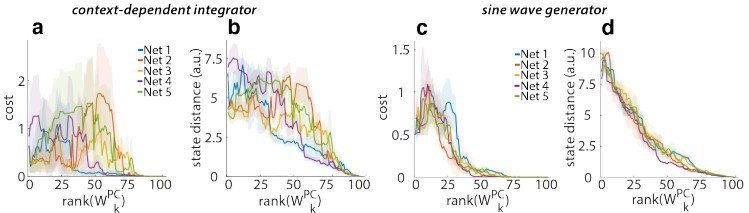

Figure 7: *Network performance for reduced-rank weight-matrices based on high-variance dimensions.* (**a**) Network output cost (Eq. 3) of networks with reduced-rank weight matrices $\mathbf{W}_k^{PC}$ for $k = 1 : N$ (Eq. 5). (**b**) State distance between trajectories in the full-rank network and in networks with reduced-rank weight matrix $\mathbf{W}_k^{PC}$. (**c-d**) Same as (**a-b**) for 5 example networks trained on sine wave generation. (**a-d**) Shown for 5 networks each; shaded area: $mad$ over trials; Network output cost obtained with internal and input noise, state distance without any noise. The network performance shows large jumps at specific ranks that differ across networks (similar to Fig. 1 e, j).

### A.3.3 Alignment between global operative dimensions and high-variance dimensions

The large difference between operative and high-variance dimensions also becomes apparent when comparing their pairwise alignment to each other. While the subspace angles between the first few global operative dimensions to the first few high-variance dimensions show a weak alignment, the remaining dimensions are almost orthogonal to each other (Fig. S8; $subspace\ angle = acos(|\mathbf{q}_i \cdot PC(\mathbf{W})_j|)$).

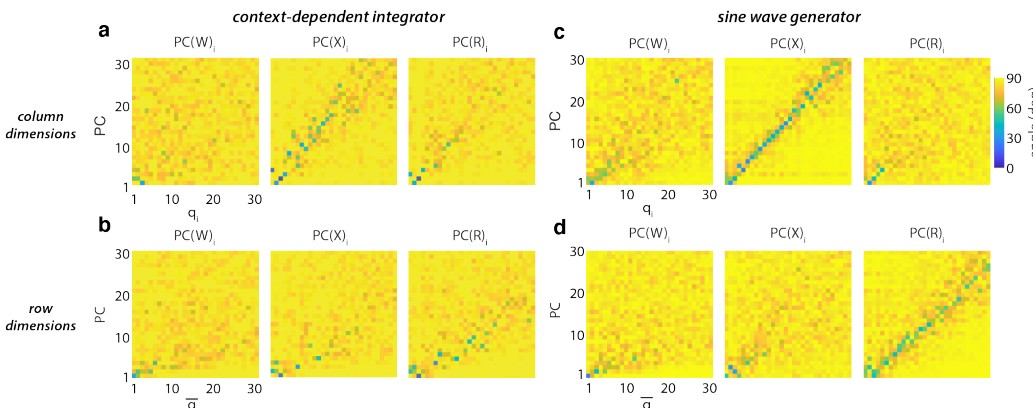

Figure 8: *Alignment of operative to high-variance dimensions.* (**a**) Subspace angle between global operative column dimensions $\mathbf{q}_i$ and PCs of $\mathbf{W}$, $\mathbf{X}$ and $\mathbf{R}$. (**b**) Same as (**a**) for global operative *row* dimensions in context-dependent integration. (**c**) Same as (**a**) for global operative *column* dimensions in sine wave generation. (**d**) Same as (**a**) for global operative *row* dimensions in sine wave generation. (**a** - **d**) Average over 20 networks per task.

Interestingly, the global operative column dimensions show a stronger alignment with the linear network activity $PC(\mathbf{X})$ (Fig. S8a, c) and the global operative row dimensions with the non-linear network activity $PC(\mathbf{R})$ (Fig. S8b, d). However, despite their partial alignment the PCs of the network activity are not describing the functionally relevant subspace as accurately as the global operative dimensions. To assess this, we construct reduced-rank approximations of $\mathbf{W}$ using $PC(\mathbf{X})_i$:

$$\mathbf{W}_k^{PC(X)} = \Sigma_{i=1}^k PC(\mathbf{X})_i (PC(\mathbf{X})_i^T \mathbf{W}) \tag{17}$$

and similarly $PC(\mathbf{R})_i$:

$$\mathbf{W}_k^{PC(R)} = \Sigma_{i=1}^k PC(\mathbf{R})_i (PC(\mathbf{R})_i^T \mathbf{W}) \tag{18}$$

Analogously for row dimensions based on Eq. 15. The networks require a larger number of $PC(\mathbf{X})_i$ or $PC(\mathbf{R})_i$ than global operative dimensions in $\mathbf{W}$ to achieve the same performance level (Fig. S9).

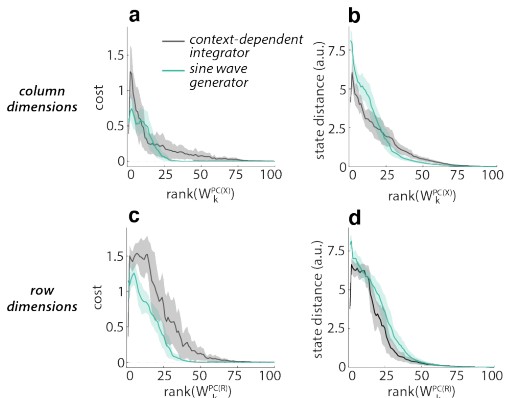

Figure 9: *Network performance for reduced-rank RNN based on PCs of network activity $\mathbf{X}$ and $\mathbf{R}$.* (**a**) Network output cost of networks with reduced-rank weight matrix $\mathbf{W}_k^{PC(\mathbf{X})}$ for $k = 1 : N$ (Eq. 17). (**b**) State distance between trajectories in the full-rank network and in networks with reduced-rank weight matrix $\mathbf{W}_k^{PC(\mathbf{X})}$ for $k = 1 : N$. (**c** - **d**) Same as (**a** - **b**) for removing $PC(\mathbf{R})_i$ from row dimensions of $\mathbf{W}$. (**a** - **d**) Averaged over 20 networks per task; shaded area: $mad$ over networks; Network output cost obtained with internal and input noise, state distance without any noise.

### A.3.4 Alignment to other relevant dimensions

While the operative dimensions are not well aligned with the high-variance dimensions shown above, they may be aligned with other important dimensions of the RNN. Here we consider three additional types of dimensions: (1) the input dimensions; (2) the eigenvectors of the weight matrix; and (3) the main dimensions characterizing the local linear dynamics around the chosen sampling locations.

First, we considered the alignment between the global operative dimensions and the input dimensions (Fig. S10), which we define based on various different approaches: we consider the principal components of the input weight matrix $\mathbf{B}$ (panel a; $subspace\ angle = acos(|\mathbf{q}_i \cdot PC(\mathbf{B})_j|)$); the main directions in state space effectively explored by the inputs (panel b; $subspace\ angle = acos(|\mathbf{q}_i \cdot PC(\mathbf{B}\mathbf{u}_t)_j|)\ \forall t = 1 : T$); or directly the column space of $\mathbf{B}$ (panel c; $subspace\ angle = acos(|\mathbf{q}_i \cdot column(\mathbf{B})_j|)\ \forall j = 1 : U$). Irrespective of the employed definition, all these inputs dimensions are largely orthogonal to the column operative dimensions, and only weakly aligned to a few row operative dimensions. Some alignment with the row operative dimensions can be expected, as these dimensions in $\mathbf{W}$ describe the *input* connections of the hidden units. It seems reasonable that the functionally relevant subspace of the row space in $\mathbf{W}$ - described by the global operative row dimensions - is at least partially aligned with the task input dimensions, as these mediate the inputs that are crucial drivers of the network activity while performing the task.

Second, we consider the alignment between the global operative dimensions and the right and left eigenvectors of $\mathbf{W}$ (Fig. S11; $subspace\ angle = acos(|\mathbf{q}_i \cdot right/left\ eigenvector\ (\mathbf{W})_j|)$). Both eigenvectors are at most weakly aligned to the global operative dimensions, emphasizing that our approach retrieves dimensions that may not be directly identifiable based on the weight matrix alone.

Third, we considered the alignment between a local operative dimensions estimated at a particular state-space location and the linearized RNN dynamics $\mathbf{A}$ at that location (linearized at sampling locations on condition average trajectory, see Eq. 24 and 25 for definition of $\mathbf{A}$). In Fig. S12 we characterize the linear dynamics through the $PC$ of of $\mathbf{A}$, which are not aligned with with respective first local operative dimensions at any sampling location (large subspace angles; $subspace\ angle = acos(|\mathbf{d}_{1,j} \cdot PC(\mathbf{W})_j|)$). Likewise, we failed to find any alignment between the local operative dimensions and the right and left eigenvectors of $\mathbf{A}$ at any sampling location (not shown).

The mismatch between operative dimensions and linear dynamics might at least partly reflect our definition of operative dimensions, which is based entirely on the contribution of the recurrent dynamics $\mathbf{W}r$ while discarding the decay term $-\mathbf{x}dt/\tau$ (see Eq. 1). The linearized dynamics, on the other hand, includes contributions from both terms. Hence, an interesting extensions of our presented definition of operative dimensions would additionally consider the decay term to define $\overline{\Delta f}$ and remove $\mathbf{a}$ from the decay term similar to as from $\mathbf{W}$ (Eq. 6). However, such a formulation would make the resulting operative dimensions harder to interpret and more work is required to gain more insights into such alternative definitions of operative dimensions.

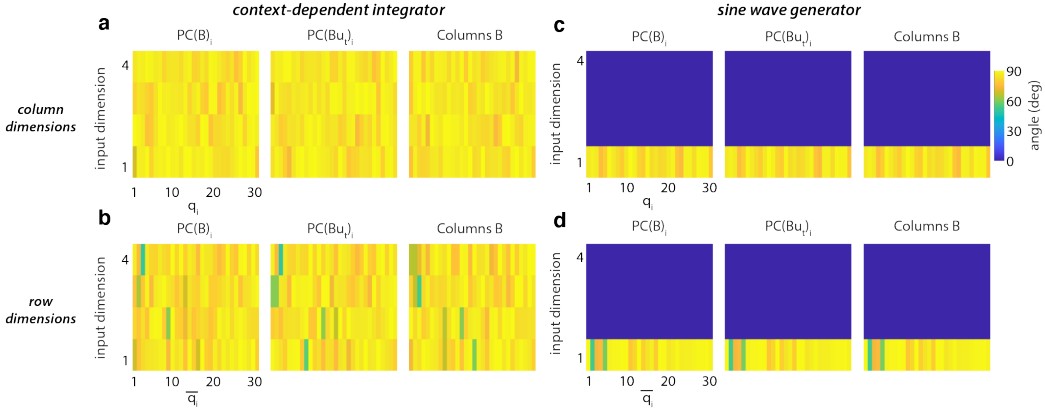

Figure 10: *Alignment between global operative dimensions and network input dimensions.* (**a**) Subspace angle between global operative column dimensions $\mathbf{q}_i$ and PCs of $\mathbf{B}$, the PCs of $\mathbf{Bu}_t$, and the columns of $\mathbf{B}$. (**b**) Same as (**a**) for global operative *row* dimensions in context-dependent integration. (**c**) Same as (**a**) for global operative *column* dimensions in sine wave generation. (**d**) Same as (**a**) for global operative *row* dimensions in sine wave generation. (**a** - **d**) Average over 20 networks per task.

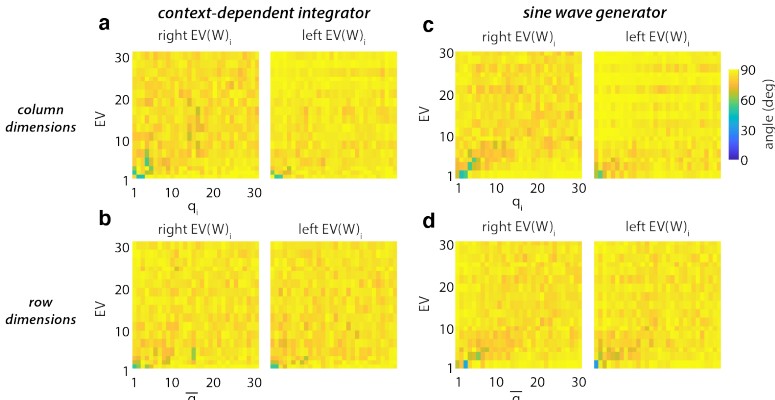

Figure 11: *Alignment between global operative dimensions and eigenvectors of $\mathbf{W}$.* (**a**) Subspace angle between global operative column dimensions $\mathbf{q}_i$ and right and left eigenvectors of $\mathbf{W}$. (**b**) Same as (**a**) for global operative *row* dimensions in context-dependent integration. (**c**) Same as (**a**) for global operative *column* dimensions in sine wave generation. (**d**) Same as (**a**) for global operative *row* dimensions in sine wave generation. (**a** - **d**) Average over 20 networks per task.

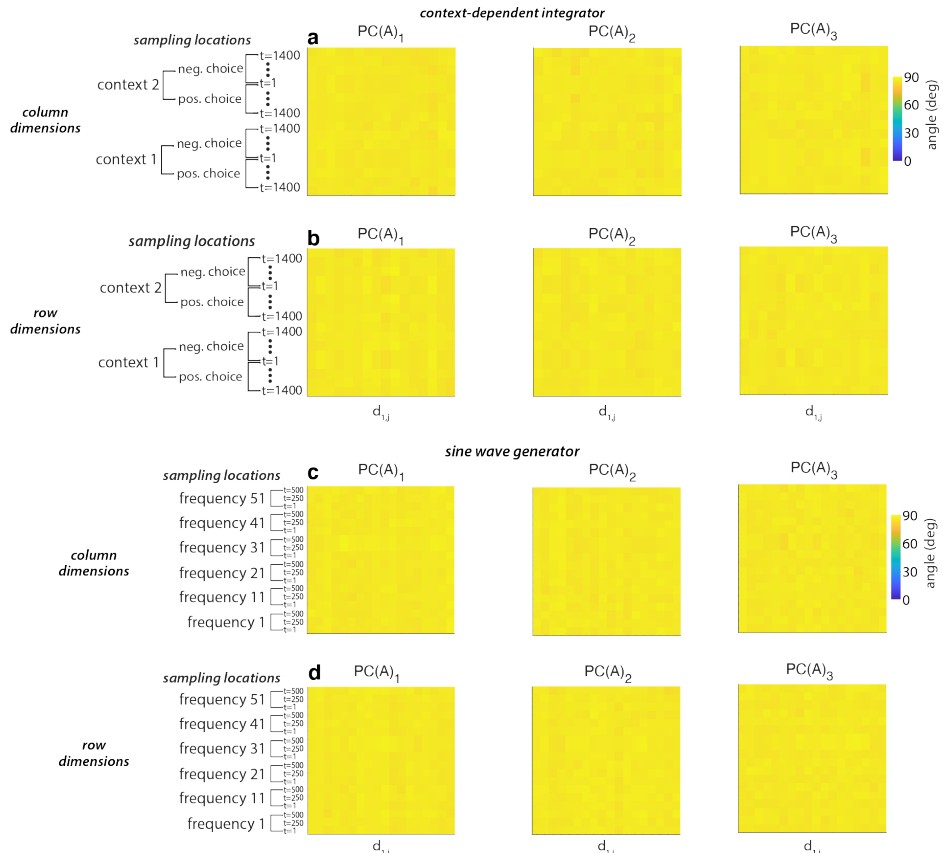

Figure 12: *Alignment between local operative dimensions and linearized dynamics $A$.* (**a**) Subspace angle between the first local operative column dimensions $\mathbf{d}_{1,j}$ and the first three PCs of $\mathbf{A}$ compared over sampling locations. (**b**) Same as (**a**) for global operative *row* dimensions in context-dependent integration. (**c**) Same as (**a**) for global operative *column* dimensions in sine wave generation. (**d**) Same as (**a**) for global operative *row* dimensions in sine wave generation. (**a** - **d**) Average over 20 networks per task.

### A.3.5 Importance of input dimensionality

To understand if the dimensionality of the global operative dimensions is related to the dimensionality of the network input we study operative dimensions in networks with systematically varied number of inputs. Specifically, we train RNNs ($N = 100$) on an extended version of the context-dependent integration task in which the networks are trained to distinguish between up to 9 contexts simultaneously ($U = 1 : 9$, $Z = 1$, Fig. 13a). We trained 5 networks for every $U$ ($U = 1 : 9$, $Z = 1$) and obtained the global operative dimensions for each of them.

In networks with higher number of inputs the dimensionality of the global operative dimensions is generally higher (Fig. 13c, f) and they require a larger number of dimensions to perform the task with the same performance (Fig. 13d,e,g,h). The overall dimensionality of the network activities also increases with $U$ (Fig. 13b) which further demonstrates the tight link between the dimensionality of the network activity and operative dimensions. Note that the dimensionality of $\mathbf{W}$ remains roughly the same over all $U$ (not shown).

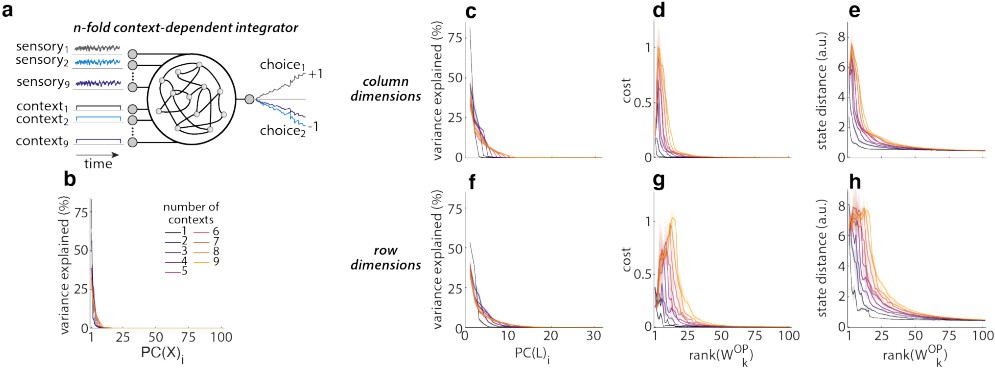

Figure 13: *Operative dimensions and input dimensionality.* (**a**) Task schematic for n-fold context-dependent integrator networks trained to select and integrate the relevant input out of 1-9 sensory inputs simultaneously. (**b**) Variance explained (in activity space) by individual PCs of the network activity $\mathbf{X}$ over all input conditions. (**c**) Rank of global operative column dimensions, estimated with PC analysis on concatenated local operative dimensions $\mathbf{L}$ (Eq. 10 and 11). (**d**) Network output cost of networks with reduced-rank weight matrix $\mathbf{W}_k^{OP}$ for $k = 1 : N$ (Eq. 12). (**e**) State distance between trajectories in the full-rank network and in networks with reduced-rank weight matrix $\mathbf{W}_k^{OP}$. (**c-e**) Based on global operative *column* dimensions; averaged over 5 networks per number of contexts; shaded area: $mad$. Network output cost obtained with internal and input noise, state distance without any noise. (**f-h**) Same as (**c-e**) for global operative *row* dimensions.

### A.3.6 Operative dimensions over training

Operative dimensions can be estimated for the RNN at any stage of training. The global operative dimensions at a particular training stage are defined based on sampling locations $\mathbf{y}_j$ located along the condition average trajectories for that training stage. Sampling locations are placed along the trajectories equally spaced in time, as described in section 2.2.1. For all training stages, a low-dimensional subspace in $\mathbf{W}$ can be defined which is sufficient to achieve the performance of the full-rank network at the same training stage (Fig. 14). Note that the network output cost for the full-rank networks is higher at early stages in training.

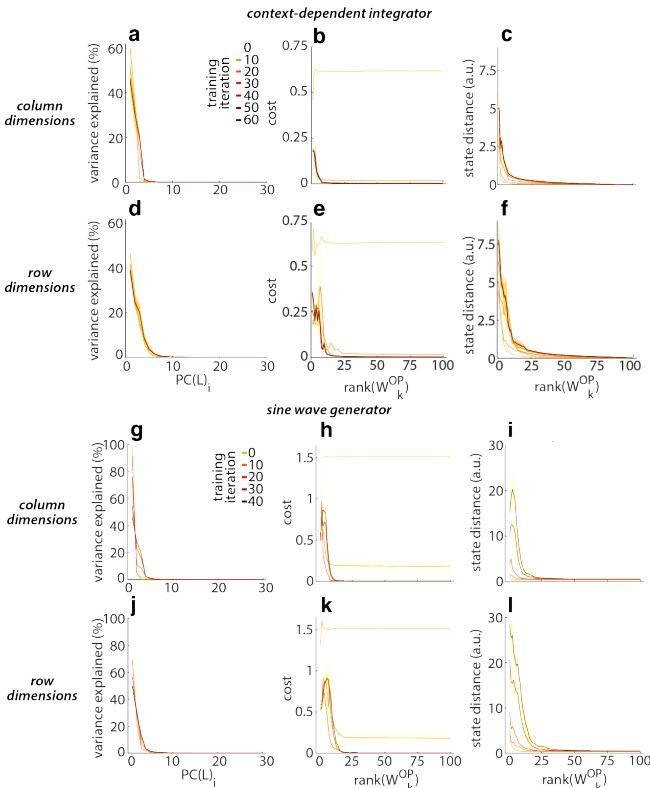

Figure 14: *Operative dimensions over training.* (**a**) Rank of global operative column dimensions, estimated with PC analysis on concatenated local operative dimensions (Eq. 10 and 11) for networks trained on context-dependent integration at different stages of training. (**b**) Network output cost of networks with reduced-rank weight matrix $\mathbf{W}_k^{OP}$ for $k = 1 : N$ (Eq. 12) trained on context-dependent integration at different stages of training. (**c**) State distance between trajectories in the full-rank network and in networks with reduced-rank weight matrix $\mathbf{W}_k^{OP}$ trained on context-dependent integration at different stages of training. (**d**-**f**) Same as (**a**-**c**) for global operative *row* dimensions in context-dependent integration networks. (**g**-**i**) Same as (**a**-**c**) for global operative *column* dimensions in sine wave generation networks. (**j**-**l**) Same as (**a**-**c**) for global operative *row* dimensions in sine wave generation networks. (**a**-**l**) Averaged over 5 networks per training iteration; shaded area: $mad$ over networks; Network output cost obtained with internal and input noise, state distance without any noise.

### A.3.7 Operative dimensions for different network types

To illustrate the general applicability of our definition of operative dimensions, we also estimated operative dimensions for the following three alternative types of RNNs:

**bias, x0.** We extended the standard RNN equation (Eq. 1) with trainable parameters for bias of hidden units $\mathbf{b}_h \in \mathbb{R}^N$, bias for output units $\mathbf{b}_z \in \mathbb{R}^Z$ and initial conditions per context$_j$ $\mathbf{x0} \in \mathbb{R}^{N \times J}$ for $J = 2$ in the context-dependent integration network.

$$\tau \dot{\mathbf{x}}_t = -\mathbf{x}_t + \mathbf{W}\mathbf{r}_t + \mathbf{B}\mathbf{u}_t + \mathbf{b}_r + \boldsymbol{\sigma}_t \tag{19}$$

where $\mathbf{x}_{t=0}$ is set to $\mathbf{x0}_j$ for trials of context$_j$. The network output is defined as:

$$\mathbf{z}_t = \mathbf{Y}\mathbf{r}_t + \mathbf{b}_z \tag{20}$$

**relu, bias, x0.** Same as Eq. 19 and 20, but replacing the $tanh$ with the $relu$ activation function for $\mathbf{r}_t$:

$$r_t(i) = max(0, x_t(i)) \; \forall i = 1 : N \tag{21}$$

**dale's law, relu, bias, x0.** Same as Eq. 19, 20, 21 and with the additional constraint on $\mathbf{W}$ to respect Dale's law which constrains hidden units to either act purely excitatory or inhibitory. Here we set 80% of the hidden units to be excitatory, 20% to be inhibitory (implementation inspired by [43]):

$$\mathbf{W} = \mathbf{W}^{rec}\mathbf{D} \tag{22}$$

with $W^{rec}(i, j) = max(0, W(i, j))$, with $i = 1 : N$, $j = 1 : N$ and a diagonal matrix $\mathbf{D} \in \mathbb{R}^{N \times N}$ defined as:

$$D(i,j) = \begin{cases} 1, & \text{if } j = i \land \text{ unit}_i \text{ is excitatory} \\ -1, & \text{elseif } j = i \land \text{ unit}_i \text{ is inhibitory} \\ 0, & \text{else} \end{cases} \tag{23}$$

To ensure convergence during training, in this last RNN type $\mathbf{W}$ was initialized with a Gamma distribution ($\mathbf{W}_0 = \mathcal{G}(2, 0.1/2)$).

We trained these alternative RNN types on the context-dependent integrator task. We find that also in these RNNs the inferred operative dimensions successfully identify a low-dimensional functional subspace of the connectivity that is sufficient to solve the task (Fig. 15).

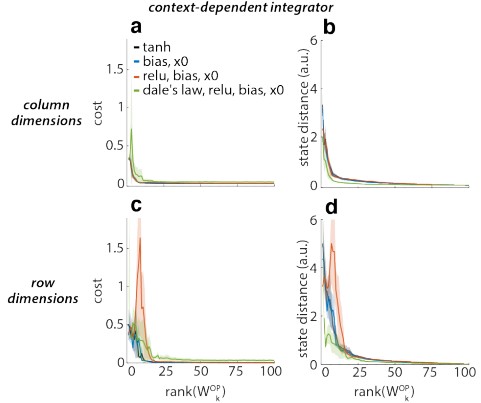

Figure 15: *Operative dimensions for various network types.* (**a**) Network output cost of networks with reduced-rank weight matrix $\mathbf{W}_k^{OP}$ for $k = 1 : N$ (Eq. 12) for different network types (section A.3.7). (**b**) State distance between trajectories in the full-rank network and in networks with reduced-rank weight matrix $\mathbf{W}_k^{OP}$ for different network types (section A.3.7). (**c-d**) Same as (**a-b**) for global operative row dimensions. (**a-d**) Averaged over 5 networks per network type; shaded area: $mad$ over networks.

### A.3.8 Operative dimensions applied to sequential MNIST

To validate our definition of operative dimensions to a task more closely related to current AI applications, we estimated operative dimensions on networks trained on sequential MNIST [44]. In this task, the RNN is trained to classify hand-written digits when the individual pixels of each image are provided sequentially over time as an input to the network. For simplicity, the RNN equations and training are analogous to those we employed for the other, simpler tasks considered above (see Eq. 1). Here we increased the size of the RNN's hidden layer from 100 to 200 units ($N = 200$), while the input is 1-dimensional ($U = 1$). This architecture does not achieve state-of-the-art performance on this task.

The properties of high-variance dimensions $PC(\mathbf{W})$ for sequential MNIST are similar to those in the simpler tasks above (Fig. 16a-d). The network activity $\mathbf{X}$ is low-dimensional throughout training while the underlying weight matrix $\mathbf{W}$ is high-dimensional (Fig. 16a, b; shown for 1 representative example network). Furthermore, sequentially removing $PC(\mathbf{W})_i$ from W would imply that the RNN requires more than 175 (out of N=200) dimensions in $\mathbf{W}$ to achieve the full-rank classification accuracy (Fig. 16a, b; results obtained on test set; full-rank classification accuracy=94% on training and test set).

To identify a functionally relevant subspace in $\mathbf{W}$ we proceeded as above. We collected the local operative dimensions at $P = 3950$ sampling locations that were equally spaced in time along a random subset of trials (at every 10th time step along 50 randomly selected trials of the training set). Placing sampling locations along the condition average trajectories (averaged over all trials of the same output class) yielded slightly worse performance, most likely due to a lower number of possible sampling locations. Overall, sequential MNIST required a higher number of sampling locations than the simpler tasks presented above [19, 20].

The resulting operative dimensions reveal that the RNN trained on sequential MNIST requires only 58 dimensions (out of 200) to perform the task with the original classification accuracy (here we define *original classification accuracy* as 95% of the accuracy of the corresponding full-rank RNN; Fig. S16; averaged over 10 networks). The operative dimensions thus identify a functionally relevant subspace that is of substantially lower dimensionality than the full-rank weight matrix $\mathbf{W}$ (results for column dimensions). Further analysis using output-class-specific operative dimensions might reveal valuable insights into the computation implemented by these networks.

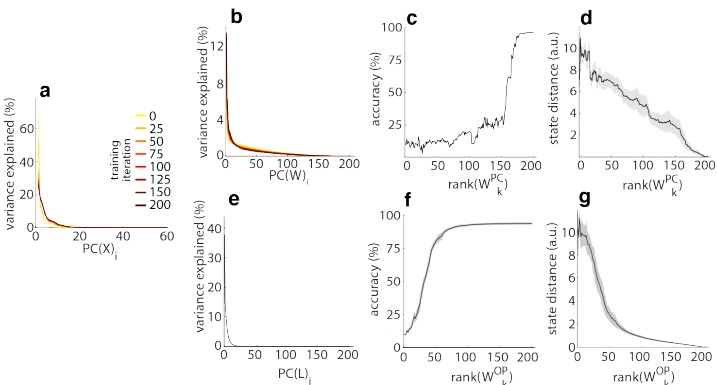

Figure 16: *Operative dimensions in sequential MNIST.* (**a**) Variance explained (in activity space) by individual PCs of the network activity $\mathbf{X}$, shown at different stages of training (test set). (**b**) Variance explained (in weight space) by individual PCs of the weight matrix $\mathbf{W}$ at different stages of training. (**c**) Network classification accuracy of networks with reduced-rank weight matrices $\mathbf{W}_k^{PC}$ for $k = 1 : N$ (Eq. 5), (**d**) State distance between trajectories in the full-rank network and in networks with reduced-rank weight matrix $\mathbf{W}_k^{OP}$. (**e**) Rank of global operative column dimensions, estimated with PC analysis on concatenated local operative dimensions (Eq. 10 and 11). (**f**) Network classification accuracy of networks with reduced-rank weight matrix $\mathbf{W}_k^{OP}$ for $k = 1 : N$ (Eq. 12). (**g**) State distance between trajectories in the full-rank network and in networks with reduced-rank weight matrix $\mathbf{W}_k^{OP}$. (**a-d**) 1 representative network. (**e-g**) Based on global operative *column* dimensions and averaged over 10 networks per task; shaded area: $mad$.

### A.3.9 Local, linear dynamics in reduced-rank RNN using operative dimensions

Computations in RNNs can often be understood by analyzing linear approximations of the dynamics around fixed points or slow points [20]. Here we ask how well the reduced-rank approximations we derived can approximate the local linearized dynamics of the full-rank networks. We find that the operative dimensions of the connectivity are sufficient to reproduce the dominant local, linear dynamics in the full-rank RNNs, reinforcing the finding that the reduced-rank RNNs capture the key computations of the full-rank or closest reduced-rank system.

Here we study the local, linear dynamics in the context-dependent integrator and the sine wave generator by linearizing around slow points of the full-rank RNN $\mathbf{x}^* \in \mathbb{R}^N$ located on the condition average trajectories [20, 19]. We obtain a linear system approximation $\mathbf{A} \in \mathbb{R}^{N \times N}$:

$$A(i,j) = -\delta(i,j) + W(i,j)tanh'(x^*(j)) \tag{24}$$

with $i = 1 : N$, $j = 1 : N$ and

$$\delta(i,j) = \begin{cases} 1, & \text{if } i = j \\ 0, & \text{otherwise} \end{cases} \tag{25}$$

Similarly, we obtain the linear system approximations $\mathbf{A}_k^{OP}$ for the reduced-rank RNN using $\mathbf{W}_k^{OP}$ with $k = 1 : N$ (Eq. 12):

$$A_k^{OP}(i,j) = -\delta(i,j) + W_k^{OP}(i,j)tanh'(x^*(j)) \tag{26}$$

with $i = 1 : N$, $j = 1 : N$.

These linear system approximations $\mathbf{A}_k^{OP}$ are then studied by analyzing their eigenvalue decomposition:

$$\mathbf{A}_k^{OP} = \sum_{i=1}^{N} \mathbf{b}_{i,k} \lambda_{i,k} \mathbf{e}_{i,k}^T \tag{27}$$

with $\mathbf{b}_{i,k} \in \mathbb{R}^N$ as the i-th right eigenvector, $\mathbf{e}_{i,k} \in \mathbb{R}^N$ as the i-th left eigenvector and $\lambda_{i,k}$ the i-th eigenvalue of $\mathbf{A}_k^{OP}$ with rank $k$. The full-rank $\mathbf{A}_{k=N}^{OP}$ consists of one dominant eigenvector ($\lambda_{1,k=N} \approx 0$) with the remaining modes fast decaying ($\lambda_{i,k=N} < 0, \forall i = 2 : N$, as described in [19]). To compare the eigenvectors and eigenvalues of the reduced-rank systems $\mathbf{A}_k^{OP}$ to each other we have to ensure a consistent sorting of their values across linearized systems $\mathbf{A}_k^{OP} \forall k = 1 : N$. Therefore we sorted the eigenvalues of the full-rank system $\mathbf{A}_{k=N}^{OP}$ in descending order of the absolute eigenvalues and then used matlab's *eigenshuffle* (https://www.mathworks.com/matlabcentral/fileexchange/22885-eigenshuffle) to sort the remaining reduced-rank systems to be as similar as possible to the full-rank system.

To measure the similarity between the full-rank and reduced-rank linear dynamics we considered the following quantities (Fig. S17):

$$angle\ to\ full-rank\ right\ EV = acos(|(\mathbf{b}_{i,k=N}^T \mathbf{b}_{i,k})|)$$

$$angle\ to\ full-rank\ left\ EV = acos(|(\mathbf{e}_{i,k=N}^T \mathbf{e}_{i,k})|)$$

$$\Delta\ to\ full-rank\ eigenvalue = |\lambda_{i,k=N} - \lambda_{i,k}|$$

Note that the linearization is performed at the location of slow points $\mathbf{x}^*$ of the full-rank RNN, which are not necessarily slow points of the dynamics in the reduced-rank RNN. This implies that in the reduced-rank networks the linearized dynamics can be expected to approximate the non-linear dynamics less well than in the full-rank network (or only at some distance from $\mathbf{x}^*$, see [20]). Despite this limitation, we find that the inferred dominant linear dynamics is largely preserved in the reduced-rank RNN. In reduced-rank RNN based on only the first few global operative dimensions, the first right eigenvector, left eigenvector and eigenvalue remain very close to their original values in the full-rank RNN; the fast decaying modes instead require the full-rank weight matrix to be retrieved (Fig. S17).

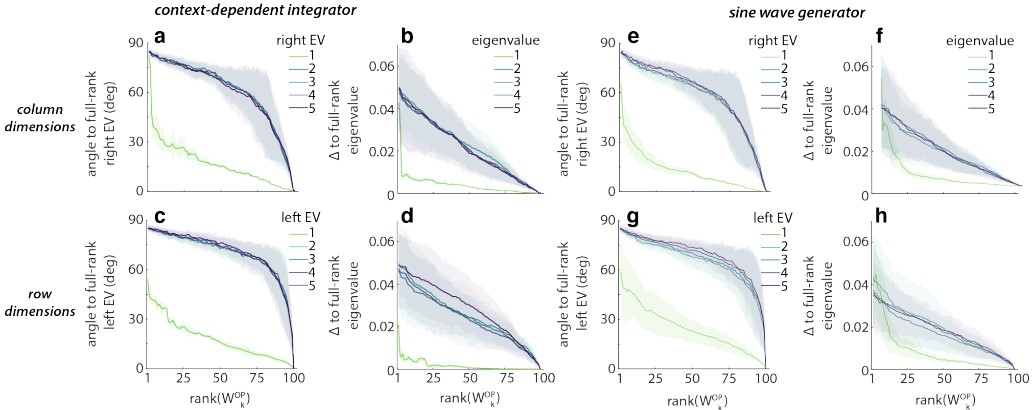

Figure 17: *Local, linear dynamics in reduced-rank networks.* (**a**) Subspace angle between the right eigenvectors of full-rank RNN and the right eigenvectors of reduced-rank RNN with $\mathbf{W}_k^{OP}$ for $k = 1 : N$. (**b**) Absolute difference between the eigenvalues of full-rank RNN and the eigenvalues of reduced-rank RNN with $\mathbf{W}_k^{OP}$ for $k = 1 : N$. (**c**) Subspace angle between the left eigenvectors of full-rank RNN and the left eigenvectors of reduced-rank RNN with $\overline{\mathbf{W}}_k^{OP}$ for $k = 1 : N$ (global operative row dimensions). (**d**) Absolute difference between the eigenvalues of full-rank RNN and the eigenvalues of reduced-rank RNN with $\overline{\mathbf{W}}_k^{OP}$ for $k = 1 : N$ (global operative row dimensions). (**a** - **d**) For context-dependent integration RNN; shaded area: $mad$ over $P = 120$ sampling locations of 1 representative network. (**e** - **h**) Same as (**a** - **d**) for sine wave generator with $P = 121$.

### A.3.10 Operative dimensions for different number of sampling locations

To accurately identify the functionally relevant subspace in $\mathbf{W}$ it is crucial to define appropriate and sufficient sampling locations $\mathbf{y}_t$ to collect the local operative dimensions at. To illustrate how the inferred operative dimensions change depending on the number of sampling locations, we systematically reduced the number of sampling locations used to generate the global operative dimensions while keeping the sampling locations equally distributed over all condition average trajectories (Fig. S18). The global operative dimensions of the context-dependent integration networks are still accurate with fewer sampling locations, whereas the global operative dimensions of the sine wave generator networks generally require more sampling locations to properly capture the functionally relevant subspace in $\mathbf{W}$.

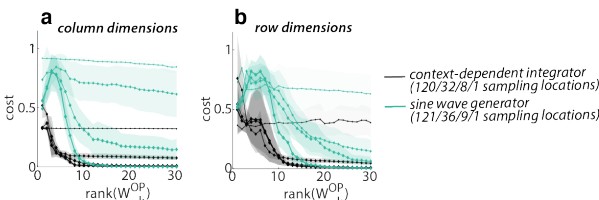

Figure 18: *Operative dimensions for different number of sampling locations.* (**a**) Network output cost of networks with reduced-rank weight matrix $\mathbf{W}_k^{OP}$ for $k = 1 : N$ (Eq. 12) for network trained on context-dependent integration using different number of sampling locations. (**b**) Same as (**a**) for global operative *row* dimensions. (**a** - **b**) Averaged over 20 networks per task; shaded area: $mad$ over networks; size of diamond-markers corresponds to indicated number of sampling locations (see legend).

### A.3.11 Single unit contributions to the functional subspace of the connectivity

Our analysis using function-specific operative dimensions (section 2.2.3) showed for the context-dependent integration network that different functional modules are implemented using distinct subspaces in $\mathbf{W}$. Functionally relevant weight subspaces are shared between choice$_1$ and choice$_2$, but not between context$_1$ and context$_2$ (Fig. 5). Here we ask if this functional specificity of operative dimensions is reflected also at the level of the connectivity of single units. Specifically, we focus on two simple properties of a given network unit, namely its effective output and total recurrent input. We define these two quantities for specific task conditions and times by exploiting the inferred operative dimensions.

We consider the effective output and total recurrent input of each unit at every sampling location $\mathbf{y}_j$ ($j = 1 : P$) separately by generating one reduced-rank approximations $\mathbf{W}_{k,j}^{OP_y}$ ($k = 1 : N$, $j = 1 : P$) per sampling location. Every $\mathbf{W}_{k,j}^{OP_y}$ consists of only the local operative dimensions defined for the respective sampling location $\mathbf{y}_j$ and thereby provides a reduced-rank approximation that is tailored to perform only specific parts of the network function, similarly to our approach in in defining function-specific operative dimensions (section 2.2.3). Using these $\mathbf{W}_{k,j}^{OP_y}$ allows us to isolate the input and output of each unit that is functionally relevant to solve specific network functions, i.e. to reproduce activity at specific times and conditions.

We define the total effective output that unit $i$ sends to all other hidden units as the norm of the weight matrix column $i$ scaled by the network activity of unit $i$ at sampling location $\mathbf{y}_j$ ($tanh(y_t(i)) = r_t(i)$):

$$\text{total effective output unit}_\text{i} = \sqrt{\Sigma_{h=1}^{N}(W_{k,j}^{OP_y}(h,i)r_t(i))^2} \tag{28}$$

This total effective output is defined separately for each sampling location $\mathbf{y}_j$. Here $k = 1$, as the operative column dimensions at every location are always rank 1 ($k = 1$; see section A.2.3).

Similarly, we define the total *recurrent* input received by each unit $i$ from all other hidden units as the dot product of the weight matrix row $i$ with the network activity $\mathbf{r}_t = tanh(\mathbf{y}_t)$.

$$\text{total recurrent input unit}_\text{i} = \Sigma_{h=1}^{N}W_{k,j}^{OP_y}(i,h)r_t(h) \tag{29}$$

Again, this recurrent input is defined separately for each sampling location $\mathbf{y}_j$. Here $k$ was set to include only functionally relevant local operative row dimensions for which $\Delta f > 10^{-6}$ ($k \approx 10$).

These total recurrent inputs and effective outputs are shown in Fig. S19a, d) for each unit (x-axis) and time/condition (y-axis). This plot does not reveal any obvious structure. Specifically, we find no evidence that particular units are contributing to the functional inputs or outputs preferentially in particular conditions but not others (e.g. context or choice).

In addition to the above unit properties, which combine information about the reduced-rank weight matrix $\mathbf{W}_{k,j}^{OP_y}$ and the network activities $\mathbf{r}_t$, we also considered simpler properties based only on $\mathbf{W}_{k,j}^{OP_y}$, i.e. only on the units' connectivity. Specifically, we analyzed the norm of each column and each row in the $\mathbf{W}_{k,j}^{OP_y}$, at every sampling location $\mathbf{y}_t$, as an alternative measure of the contribution of each unit to specific network functions (Fig. S19b-c, e-f). However, similar to Fig. 19a, d), these measures do not show any obvious structure over time and conditions at the level of single units.

Overall these observations suggest that all neurons are at least partially involved in creating the functionally relevant RNN dynamics at all times and in all conditions. This becomes apparent in Fig. 19a-b, d-e) as the values along every column (corresponding to the total recurrent input or effective output per unit) change abruptly between sampling locations from similar times and conditions (y-axis). However, more detailed analysis of the reduced-rank weight matrices and network activities may well reveal structure at the level of units that is not apparent from the simple properties that we analyzed here.

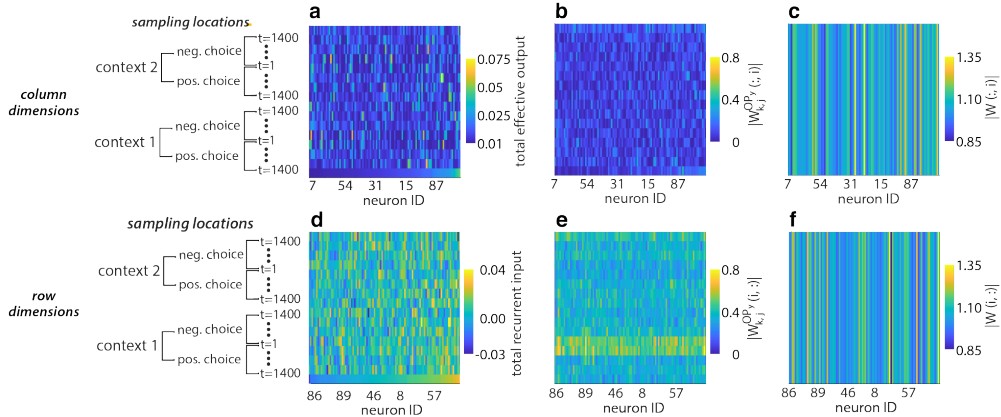

Figure 19: *Functional contribution of individual network units.* (**a**) Total effective output of each unit $i$ at sampling location $\mathbf{y}_j$ as defined in Eq. 28. (**b**) Norm of column $i$ in reduced-rank weight matrix $\mathbf{W}^{OP_y}_{k,j}$ at different sampling locations $\mathbf{y}_j$ (i corresponding to neuron ID along x-axis). (**c**) Norm of column $i$ in full-rank weight matrix $\mathbf{W}$ at different sampling locations $\mathbf{y}_j$ (i corresponding to neuron ID along x-axis). (**d**) Total recurrent input of each unit $i$ at every sampling location $\mathbf{y}_j$ as defined in Eq. 28. (**e**) Norm of row $i$ in reduced-rank weight matrix $\mathbf{W}^{OP_y}_{k,j}$ at different sampling locations $\mathbf{y}_j$ (i corresponding to neuron ID along x-axis). (**f**) Norm of row $i$ in in full-rank weight matrix $\mathbf{W}$ at different sampling locations $\mathbf{y}_j$ (i corresponding to neuron ID along x-axis). (**a-c**) sorted by values of last row in (**a**). (**d-f**) sorted by values of last row in (**d**). (**a-f**) Sampling locations subsampled, shown are locations with congruent inputs at t=1, 400, 900, 1400.

### A.3.12 Alignment between global operative column and row dimensions

The global operative column and row dimensions both span a low-dimensional subspace in the weight matrix $\mathbf{W}$ which is sufficient for the RNNs to perform the task (Fig. 3b-g). However, the two subspaces show little similarity. In the context-dependent integration network, only the first global operative column dimensions are weakly aligned to the first operative row dimensions. The remaining dimensions show no alignment to each other (Fig. S20a). Similarly in the sine wave generator networks, the global operative column and row dimensions show little similarity (Fig. S20b).

Overall, operative column and row dimensions provide complementary insights into RNN computations. In broad terms, the column dimensions in $\mathbf{W}$ describe the *output connections* of each hidden unit, whereas the row dimensions describe the *input connections* of each hidden unit. The respective operative dimensions in turn identify the functionally relevant subspaces in the network connectivity. If these subspaces of the network connectivity are interpreted as subspaces in the network activity, they might provide a tool to compare each unit's functionally relevant input and output subspaces, i.e. to determine how the activity of a given hidden unit is shaped by, and how it influences, the activity in the remainder of the network.

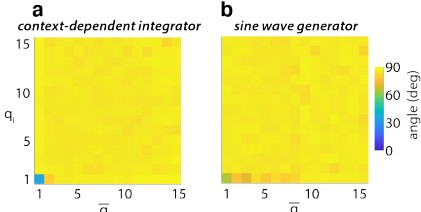

Figure 20: *Alignment between global operative column and row dimensions.* (**a**) Subspace angle between global operative column dimensions $\mathbf{q}_i$ and global operative row dimensions $\overline{\mathbf{q}}_i$ for context-dependent integration networks. (**b**) Same as (**a**) for sine wave generation networks. (**a-b**) Averaged over 20 networks.

### A.3.13 Alignment of operative dimensions over sampling locations

The global operative dimensions are a combination of the local operative dimensions of all sampling locations. To illustrate the large differences between local operative dimensions across sampling locations, here we show how the local operative dimensions from different sampling locations are aligned to each other. We find that local operative dimensions tend to be similar if their sampling locations are close to each other in state space. However, more distant sampling locations generally yield almost orthogonal local operative dimensions (Fig. S21b, e). Similarly, we test the alignment between the local operative dimensions at a particular sampling location and the global operative dimensions. We find that the global operative dimensions are not preferentially aligned to any local operative dimensions defined at particular sampling location, but rather are partially aligned to the local operative dimensions from all sampling locations (Fig. S21c, f, i, l).

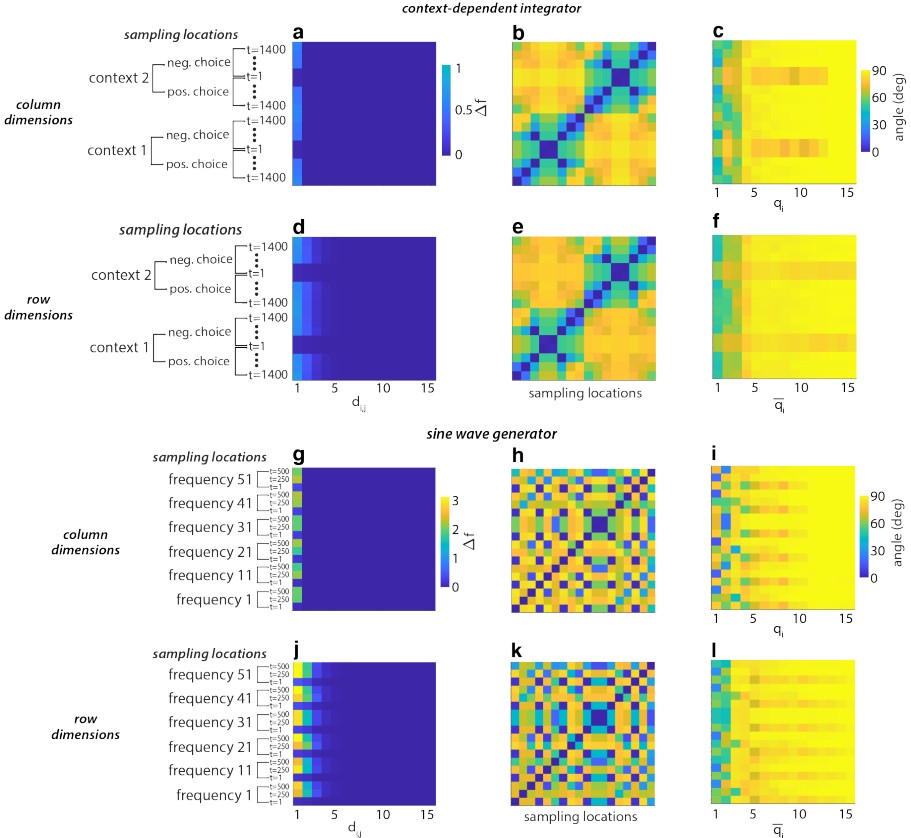

Figure 21: *Alignment of operative dimensions over sampling locations.* (**a**) $\Delta f$ for local operative column dimensions of context-dependent integrator. The subspace of local operative column dimensions is 1-dimensional at all sampling locations. (**b**) Pairwise subspace angle of first local operative column dimensions across sampling locations. Local operative column dimensions gradually change over state space, with closer sampling locations yielding more similar local operative column dimensions. (**c**) Subspace angle between the first local operative column dimension at each sampling location and the global operative column dimensions. The first few global operative column dimensions are partially aligned with local operative column dimensions from most sampling locations. (**d-f**) Same as (**a-c**) for operative *row* dimensions for context-dependent integrators. (**g-i**) Same as (**a-c**) for operative *column* dimensions for sine wave generators. (**j-l**) Same as (**a-c**) for operative *row* dimensions for sine wave generators. (**a-f**) Sampling locations are sorted based on the spatial proximity to each other, moving along the line attractor over time in each context. (**g-l**) Sampling locations are sorted based on the input frequency and then the time along the respective condition average trajectory, showing that local operative dimensions are not shared per frequency. (**a-l**) Averaged over 20 networks; only a subsample of all sampling locations shown; all subspace angles are computed considering only the first local operative column or row dimensions at each sampling location.