# OpenReview forum: "Operative dimensions in unconstrained connectivity of recurrent neural networks"
_NeurIPS.cc/2022/Conference — NeurIPS 2022 Accept_

### Official Review · Reviewer_QY7N · 2022-07-10

**Rating:** 6
**Confidence:** 5
**Soundness:** 3 good
**Presentation:** 4 excellent
**Contribution:** 3 good

**Summary:**

The authors study how unconstrained training of recurrent neural networks results in structured synaptic connectivity. Specifically, they define a novel decomposition of the synaptic matrix that reveals this structure. In contrast to a naïve decomposition of the matrix, this novel form is relatively low-dimensional.
The new suggestion is to define operative dimensions as those directions in weight space that lead to the maximal disturbance to neural dynamics. These are defined locally, around points of interest, and then integrated into a global measure.


**Questions:**

1.	Besides reference 20, there are also relevant works in the feedforward domain. A couple of examples:
a.	Saxe, Andrew M., James L. McClelland, and Surya Ganguli. "Exact solutions to the nonlinear dynamics of learning in deep linear neural networks”
b.	Timor, Vardi, Shamir, “Implicit Regularization Towards Rank Minimization in ReLU Networks”
2.	L38-39: “… it remains unclear … findings can be generalized to RNN … not subject to such constraints”. – This sentence highlights the need to better relate to existing literature.
3.	Equation 1 and Line 57. If I understand correctly, the random noise is drawn independently for every dt. Stochastic differential equations require scaling the noise with sqrt(dt). If dt is not changed, this just amounts to an “incorrect” scaling of the noise. But, in general, it is not an exact implementation of equation 1, and could lead to problems when checking sensitivity for dt for instance.
4.	It is not entirely clear to me why the operative dimensions reveal cases similar to figure 2. If a change in W causes a trajectory to cross a separatrix, the local changes might be small, but the resulting global (t -> infty) changes could be large. The discussion in section 2.2.1 explains why a small change in local dynamics can lead to dramatic performance changes. But operative dimensions search for large local changes.
5.	L228: Another relevant reference here is Smith, Linderman and Sussillo “Reverse engineering recurrent neural networks with Jacobian switching linear dynamical systems”
6.	Is there any relation between these results and the eigenvalue spectrum of W, or of its linearized versoins? If I understand correctly, the authors show that singular value decomposition is uncorrelated to these dimensions. On the other hand (following the jacobian switching linear systems reference above, and the results in A.2.3), is there a relation to the eigenvectors of the linearized dynamics around the points used?


**Limitations:**

Yes

**Strengths And Weaknesses:**

Strengths:
This is an important problem, and there are few studies tackling it. The suggested approach is, to the best of my knowledge, novel.
Weaknesses:
A main weakness is novelty with respect to reference 20 (Schuessler et al, NeurIPS 2020). That paper tackled the same problem, and showed both numerically and analytically that the changes to connectivity (delta W) are low rank. There are certainly differences between the papers, but given the similarity, this is hardly discussed.
Another relevant reference (that is not cited) is Smith, Linderman and Sussillo “Reverse engineering recurrent neural networks with Jacobian switching linear dynamical systems”, NeurIPS 2021. In that paper, roughly speaking, trained networks are approximated by several local dynamical systems. It would seem that the analysis of A.4.3 of Vanilla might be extended to general networks by using the Jacobian at each point in phase space.
The paper also has very little analytical results – specifically, the question of why this low dimensionality develops is not addressed.

---

> ### Author Response · Authors · 2022-08-02
> **Reply to reviewer QY7N**
>
> We thank the reviewer for the constructive feedback. We address the specific questions and comments below:
>
> (1) _Weakness: Lack in novelty with respect to (Schuessler et al., NeurIPS 2020)._ \
> We agree with the reviewer that the work of Schuessler et al. is closely related to our manuscript, as it addresses the same overall question (relating structure to function in RNNs) and presents findings that are likely to be related. Moreover, our analysis of the “high-variance dimensions” of the connectivity (the principle components of W) is directly inspired by that paper (but as we show cannot in general identify a low-d functional subspace of the connectivity; Fig. 1e,j). However, our findings do not directly follow from this earlier work. Schuessler et al find that the overall weight updates (delta-W) are low-rank, both when the initial connectivity is random or set to zero. This finding alone, however, implies that the final connectivity is low-rank only when the initial connectivity is zero. A low-rank change of a high-rank (random) connectivity, instead, does not necessarily imply that the final connectivity will be low-rank. For this reason, our finding that a low-dimensional subspace of the recurrent weight matrix in unconstrained, randomly initialized RNNs is sufficient to perform the task does not follow from the analysis in Schuessler et al.. In addition, we show that the methodology proposed by Schuessler et al for identifying the relevant dimensions of the connectivity (PCA), while successful when applied to the delta-W, does not work when applied to the full connectivity in unconstrained settings, where instead operative dimensions are required. We expanded the discussion of this past work to better acknowledge the connections (Section 3).
>
> (2) _“It would seem that the analysis of A.4.3 of Vanilla might be extended to general networks by using the Jacobian at each point in phase space.”_ \
> We would like to ask the reviewer for clarification on this comment. Our manuscript does not include a section A.4.3. Also, it is not immediately clear how that section in Smith et al. relates to the reviewer’s question.
>
> (3) _Weakness: Lack of analytical results on low-dimensionality of operative dimensions._ \
> We agree with the reviewer that our findings raise a number of questions that may be addressed with analytical approaches, although analytical descriptions of unconstrained non-linear RNNs have generally proven difficult (e.g. analytical derivations in Schuessler et al. based on linear networks). Beyond the simple derivations in sections A.2.3-5, the focus of our manuscript is on defining operative dimensions, and on extensively validating them empirically in trained networks. We have further strengthened this aspect of our work in the revision, by analyzing networks trained on sequential MNIST (Supp. Fig. 15 and reply to reviewer 49zS). These novel experiments suggest that operative dimensions will be a valuable tool in relating network structure to function also in the setting of tasks more relevant for AI.
>
> (4) _Question 1, 2, 5: Relation to existing literature._ \
> Thank you for pointing out these additional relevant references, which we included in the manuscript together with additional relevant work pointed out by the other reviewers.
>
> (5) _Question 3: Scaling of random noise._ \
> In our networks we are indeed scaling the noise with sqrt(dt), as suggested. We omitted this scaling in the equation for simplicity. We updated the revised manuscript (Sections 2, A.1.1).
>
> (6) _Question 4: Relation of operative dimensions to Fig. 2._ \
> As defined in the manuscript, operative dimensions only capture contributions from the recurrent component of the dynamics (W*r), while ignoring the decay term (-x dt/tau; see also point 5 in reply to reviewer emwn). With this definition, a single, local column operative dimension (or a few row dimensions) is sufficient to capture the recurrent contribution at any particular state space location (for vanilla RNN, section A.2.3). This single local operative dimension thus explains *all* the contributions of the connectivity to the local dynamics at that location, whether those contributions are large or small.
>
> (7) _Question 6: Relation of operative dimensions and eigenvalue decomposition of W or linear dynamics._ \
> We now report the similarity of operative dimensions to other network axes, including those proposed by the reviewer, in the appendix (Section A.3.4). One main finding is that operative dimensions show little similarity to these dimensions, including the principal components of the linear dynamics. As we explain in the appendix, this mismatch may at least partly reflect our definition of operative dimensions, which is motivated by computational neuroscience, and unlike the linear dynamics discards the decay term (-x dt/tau). As mentioned in our reply to reviewer emwn, we will explore alternative definitions of operative dimensions that include the decay term.

---

> > ### Comment · Reviewer_QY7N · 2022-08-06
> > **A.4.3**
> >
> > Typo.
> > I meant A.2.3
> >
> > Now reading the other reviews, rebuttal, manuscript...

---

> > > ### Author Response · Authors · 2022-08-09
> > > **Reply to A.4.3**
> > >
> > > We thank the reviewer for this clarification, and for the useful suggestion. Indeed, key derivations in A.2.3 on local operative column dimensions can be extended to the case where the non-linear RNN dynamics is well described by the local linear approximation. In this sense, the derivations and corresponding results are more general, i.e. they apply (approximately) to any RNN architecture that results in dynamics that are locally linear, not just to vanilla RNN.
> > > We will extend the text in the appendix accordingly.

---

> > ### Comment · Reviewer_QY7N · 2022-08-09
> > **Post rebuttal**
> >
> > I read all the reviews, the authors' answers, and the relevant parts of the new manuscript.
> >
> > I am raising my score to a "weak accept".
> > While the specific approach is novel, it is far from being the first work to analyze the result of unconstrained training of RNNs. The authors do give a better context with respect to literature in the modified version.
> >
> > One point that was absent from the discussion is "how low is low?". Namely, the RNNs still require roughly 20 dimensions for a context integration task. This is in contrast to 1-2 for networks that were constrained to be low-rank (Mastroguiseppe et al), or 6-7 when examining delta-W for unconstrained networks (Schuessler et al). If part of the goal is interpretability, this weakens the impact of the paper.
> >
> > Nevertheless - my overall evaluation is that it is a novel contribution, for a relevant and timely problem, and it can advance the field.

---

> > > ### Author Response · Authors · 2022-08-09
> > > **Reply to Post rebuttal**
> > >
> > > We thank the reviewer for taking our response and the revised manuscript into account.
> > > We agree with the reviewer’s comment that the low-dimensional subspaces spanned by operative dimensions might be harder to interpret than the low-rank solutions presented by Mastrogiuseppe et al., which have yet lower dimensionality. We will mention this in the discussion.
> > > Nonetheless, we want to emphasize that the somewhat higher dimensionality of the functionally relevant connectivity subspace does not preclude two key, novel insights from our work, namely that (1) specific functions performed by the RNN can be related to specific subspaces of the connectivity (section 2.2.3); and (2) much of connectivity in unconstrained, trained RNN is irrelevant to solve the task (section 2.2.2).

---

### Official Review · Reviewer_Q6im · 2022-07-11

**Rating:** 7
**Confidence:** 4
**Soundness:** 4 excellent
**Presentation:** 4 excellent
**Contribution:** 3 good

**Summary:**

This paper presents a framework and method to analyze the connectivity of unconstrained RNN weights. They find that while recurrent weights are high-dimensional, the network dynamics that actually solve simple neuroscience tasks (i.e. ones that map stimulus inputs to one of two choices) are low-dimensional. These low-dimensional subspaces are estimated by removing different dimensions and examining the effect on activity state space. These are referred to as operative dimensions, which contain functional modules for the task. The operative dimensions can be related back to the recurrent connectivity matrix W by the construction of a reduced-rank connectivity W^{OP}. Finally, they illustrate how this framework allows one to relate different functional modules to subspaces in the weight matrix, which in essence relates function to structure.

**Questions:**

* Fig. 3C shows that network performance gets worse with the first few operative dimensions before it gets better. The authors hypothesize in lines 201-206 that the first few operative dimensions may be required at state space locations explored late in the trial, rather than at the beginning of the trial. Is there any evidence to support this?
* It is interesting that the activity of the RNN is low-dimensional even before training. Why is this?
* Any thoughts from the authors on how this might scale to stacked RNN architectures?

Suggestions
* The citations in the text are numbered, but the reference list is not. Please fix.

**Limitations:**

The authors thoroughly discuss the limitations to their approach, notably their choice to sample the average trajectory in activity state space. They also provide supplementary analyses to address some of the limitations.

**Strengths And Weaknesses:**

The paper is well-written, with sufficient detail and supporting material in the appendix. The math is straightforward and clear. The methods are certainly based on some prior work, but the general framework and findings are novel and interesting. The authors are also very thorough with additional analyses and explore nearly all the related questions that arose as I read the paper.

Their findings are significant in that they apply to unconstrained RNNs, which brings the literature closer to real-world uses of neural networks. They also could be applied to other areas in machine learning (continual learning or weight compression). And, as the authors note, the findings could also make it easier for neuroscientists to use RNNs as models of biological datasets. This paper could clearly have impact on several sub-areas.

One of the most interesting findings, in my opinion, is that the number of operative dimensions increases with with task/network complexity. It will be interesting to test this point further by independently varying stimulus complexity and output complexity. For example, how many more dimensions are required as one increases the number of nonlinearities required to produce some output? To be clear I am not suggesting that the authors conduct another analysis now -- there is already more than enough in the current paper.

---

> ### Author Response · Authors · 2022-08-02
> **Reply to reviewer Q6im**
>
> We thank the reviewer for the positive feedback and for appreciating the general applicability of operative dimensions. We are grateful for the interesting comments and questions which we address below:
>
> (1) _Question 1: Explanation of why RNN performance initially decreases when adding operative dimensions_ \
> The evidence underlying our interpretation of the initial decrease in performance when adding operative dimensions to the connectivity was indeed not clearly explained in the original manuscript. We have improved the relevant explanations in the revised manuscript (Section 2.2.2).
> The relevant evidence is shown in Figure 20c,f,i,l, where we compare the alignment between the global operative dimensions (x-axis) and the first local operative dimension at different sampling locations (y-axis). Each local operative dimension corresponds to the component of the connectivity that is most critical to implementing the recurrent dynamics at a particular location along an activity trajectory, and thus at a particular time in the trial. We find that the local operative dimensions for sampling locations early in the trial (t=0, see y-axis of Fig. 20c,f,i,l) are generally less aligned with the first few global operative dimensions compared to the local operative dimensions for sampling locations later in trial. Rather, the local operative dimensions at early trial locations (t=0) show a stronger alignment to “higher-order” global operative dimensions (e.g. q_i with i=5:11). This finding implies that a connectivity matrix built from only the first few global operative dimensions is missing the components that are most important to reproduce the dynamics occurring early in the trial.
>
> (2) _Question 2: Reason for low-dimensional network activity even before training_ \
> The network activity is low-dimensional even before training because of the network initialization. We set the spectral radius of the randomly initialized recurrent weight matrix W to be one. If the spectral radius is increased - while maintaining the same network inputs - the dimensionality of the network activity increases. Of course, increasing the dimensionality of the input would generally also lead to higher dimensional network activities.
> In the revised manuscript, we now mention the importance of the spectral radius of W at initialization to explain the low-dimensional activity before training (Section 2.1).
>
> (3) _Question 3: Operative dimensions for stacked RNN architectures_ \
> We thank the reviewer for this interesting question. Stacked RNNs can be described as one large, sparse weight matrix with 0 entries for all recurrent connections between layers. Hence, the general definition of operative dimensions and the relation between the dimensionality of the global operative dimensions (Section A.2.4-2.5) and the network activity are directly applicable. It will be interesting to explore the properties of operative dimensions in trained networks with such stacked architectures in the future. In the reply to the reviewers’ comments we have however focused on estimating operative dimensions on networks that are still single-layer networks, but that implement a more complex task (sequential MNIST; see Supp. Fig. 15 and the reply to reviewer 49zS). These novel experiments suggest that operative dimensions will be a valuable tool in relating network structure to function also in the setting of tasks more relevant for AI.
>
> (4) _Question 4: Add numbers to reference list_ \
> Thank you for pointing out this issue. We corrected the reference list in the revised manuscript.

---

### Official Review · Reviewer_emwn · 2022-07-14

**Rating:** 8
**Confidence:** 4
**Soundness:** 4 excellent
**Presentation:** 4 excellent
**Contribution:** 4 excellent

**Summary:**

This paper works towards an understanding of how weight structure influences the dynamics of recurrent neural networks (RNNs). It is shown that the principal components (PCs) of the weight matrix do not provide a good compression that preserves dynamics, instead a more dynamically-informed low-rank compression of the weights into "operative dimensions" can do so. These are defined (locally, along the attractor, similar to Lyapunov exponents) as the direction which maximizes norm of the divergence of trajectories. Compression of the W matrix using either PCs or operative dimensions is compared with a few RNN tasks used in computational neuroscience.

**Questions:**

 If the RNN dynamics are linearized, I believe that there is a direct connection to the singular vectors of the dynamical map. Maybe PC directions, in this special case, correspond to operative dimensions? This seems like an obvious connection to explore. Also, spending a bit more time in the text talking about the methods you use to optimize Eqn 8 and 9 would be helpful.

Do you see alignment of the operative dimension components with the B matrix components? The alignment (or lack thereof) to the W matrix components is explored extensively; this seems like an obvious one to also explore.

In the discussion, it would be worth discussing connections to model reduction methods, eg for PDEs and high-dim ODEs.

Preservation of attractor basin boundaries seems like something these operative dynamics clearly address, but I don't think this was discussed.

**Limitations:**

I found the authors did a good job discussing limitations and potential ethical issues.
However, I do think they could mention that their results are limited to studying very simple tasks, much simpler than the AI type tasks where RNNs are employed in practice.

**Strengths And Weaknesses:**

I found this paper to be very well-written and interesting. I didn't read the supplement due to lack of time. The way of defining operative dimensions is elegant and seems to be somewhat numerically efficient (it is tested for weight matrices with 1000s of dimensions). I liked how Fig 2 shows how these dimensions affect the attractor structure of the RNN, and I enjoyed the discussion.

The main weaknesses I found were in the tasks that were chosen. These tasks are somewhat artificial although I like their grounding in neuroscience-related tasks. For the NeurIPS audience, it would be interesting to see some whether operative dimensions offer insight into a more "AI" RNN task. Perhaps this is beyond the scope of the current paper.

I think the paper would be strengthened by exploring the connection between your ideas and more classical dynamics ideas such as Lyapunov vectors or bred vectors. You are essentially using the same idea, calculating directions of maximal growth in trajectories, except you are doing it in the weight space rather than the state space. This is interesting.

Something that could describe the low-dimensionality that is observed here is the dimensionality of the control signal u/matrix B. It could be that this low-dimensional drive forces the RNN onto an attractor of similar dimension. That could explain the patterns you observe in Fig 4. I imagine there is extensive literature that could explain this effect.

Some detailed comments:
Throughout the text you use "RNN" to refer to "recurrent neural networks" in the plural. I think you should change this to "RNNs" for the plural, which is more common.

148: "dimension of the connectivity a" suggest changing to unambiguous language "unit vector a"

150, eqn 6: maybe worth mentioning this is an orthogonal projection

283: "This past work had however not" awkward language. Also, I believe the low-rank RNN work from Mastrogiuseppe and Ostojic did relate to attractor structure.

---

> ### Author Response · Authors · 2022-08-02
> **Reply to reviewer emwn**
>
> We thank the reviewer for the thoughtful comments and valuable suggestions on additional literature to include in our discussion. We address the specific questions and comments below:
>
> (1) _Weakness: chosen tasks are somewhat artificial. Would be interesting to see application to a more "AI" RNN task._ \
> Following the suggestion by reviewer 49zS, we computed operative dimensions on vanilla RNNs trained on sequential MNIST (new Supp. Fig 15, section A.3.7). As we explain in more detail in the reply to reviewer 49zS, these novel experiments suggest that operative dimensions will be a valuable tool in relating network structure to function also in the setting of tasks more relevant for AI.
>
> (2) _Explore connection between operative dimensions and classical dynamic ideas (e.g. Lyapunov vectors, bred vectors)._ \
> Thank you for this suggestion. We agree that these approaches to characterizing dynamics are closely related to our definition of operative dimensions, as both rely on a local analysis of the dynamics around explored trajectories. We now point out this connection in the discussion (Section 3). It would be very interesting to explore these connections in more detail in the future.
>
> (3) _Relation of low-dimensionality to control signal u/matrix._ \
> This is an interesting suggestion. However, to the best of our knowledge, a comprehensive understanding of the exact factors determining the dimensionality of the dynamics in trained RNNs is currently lacking. The dimensionality of the control signal/inputs can be expected to be an important factor (e.g. Gao, P., & Ganguli, S., 2015) although in general not the only one. Indeed, RNNs that receive high-dimensional inputs can nonetheless generate low-dimensional dynamics (e.g. Maheswaranathan et al., 2019). On the other hand, reservoir computing networks can generate high-dimensional dynamics even when driven with low-dimensional inputs (e.g. H. Jaeger, 2003; Maass et al., 2002). Our result that the dimensionality of activity in the N-fold sine-wave generator increases with N (Fig. 4b) could further be interpreted as being driven by the dimensionality of the output. Lacking a complete understanding of how these factors interact, we have added a discussion of this point to the appendix (Section A.2.4).
>
> (4) We thank the reviewer for the 4 “detailed comments” and adjusted the manuscript accordingly.
>
> (5) _Questions 1 and 3: Relation between operative dimensions and singular vectors of dynamical map and network inputs (B/u_t)_ \
> To address this question, we now report the similarity of operative dimensions to important network axes in the appendix (Section A.3.4).
> The main empirical findings are that (i) operative dimensions show little similarity to the principal components or the eigenvectors of the linearized system approximation; (ii) the global operative row (but not column) dimensions are weakly aligned with the input dimensions (in B/u). The row dimensions in W correspond to the “input” connections of the hidden units (Appendix section A.3.11) and thus the functionally relevant input subspace might be expected to be at least partially overlapping with the input subspaces, as the network relies on these inputs to perform the tasks.
> The mismatch between operative dimensions and linear dynamics might however at least partly reflect our definition of operative dimensions, which is based entirely on the contribution of the recurrent dynamics (W*r) while discarding the decay term (-x dt/tau). The linearized dynamics, on the other hand, includes contributions from both terms.
> Our definition of operative dimensions is motivated by approaches from computational neuroscience, whereby the decay term is considered to reflect an intrinsic property of the neuron, which is different in nature from the contribution from recurrent (and input) connections from other neurons (i.e. RNN units). Operative dimensions could be defined differently, by including the decay term into the connectivity matrix W as a self–inhibition term. We now mention this point in the appendix (Section A.3.4). We will explore this alternative approach and determine whether it leads to a better alignment between operative dimensions and linear dynamics.
>
> (6) _Question 2: Optimization methods._ \
> We perform the optimization with a built-in matlab function implementing a quasi-newton optimization algorithm (additional details now in section A.2.7). We are further in the process of porting our code to Python.
>
> (7) _Question 4 and 5: Connection with model reduction methods_ \
> Thank you for this suggestion, we now discuss the relation to model reduction approaches in the discussion (Section 3).
>
> (8) _Limitations: simple tasks compared to AI tasks_ \
> As mentioned above, we now also trained RNN on sequential MNIST (point 1). In addition, we modified the discussion to mention the potential limitations of our findings on simple tasks (Section 3).

---

> > ### Comment · Reviewer_emwn · 2022-08-08
> > **Thanks**
> >
> > I appreciate the responses. I am leaving my scores the same.
> >
> > Re point (3), dim of control signal: You could compare different control dimensions in an artificial task to easily see if there is a trend. I don't think you have to do this for this paper, necessarily, but it would be very easy to test. The dimension of activity is certainly related to Lyapunov dimension, as is well-known. But even chaotic systems, in certain regimes, can be entrained to control signals.

---

> > > ### Author Response · Authors · 2022-08-09
> > > **Reply to "Thanks"**
> > >
> > > We thank the reviewer for this interesting suggestion. We will test this suggestion by implementing an N-fold context-dependent integrator, which will allow us to vary the number of inputs (N sensory inputs and N context inputs) while keeping the dimensionality of the output fixed (1 dimensional, i.e. the integral of the contextually relevant sensory input). As suggested by the reviewer, the number of required operative dimensions will likely increase with N.

---

### Official Review · Reviewer_49zS · 2022-07-15

**Rating:** 7
**Confidence:** 3
**Soundness:** 3 good
**Presentation:** 3 good
**Contribution:** 2 fair

**Summary:**

The paper presents the operative dimensions -- a novel, more effective tool (e.g. compared to PCA of W, X or R) to characterize the functionally relevant subspaces of RNN weights. The tool is well designed (Sec 2.2.1), easy to compute (appendix), and able to identify previously unknown properties of RNNs, e.g. the existence of low-dimensional functional subspaces (Sec 2.2.2) and the weight-space functional modules (Sec 2.2.3).

**Questions:**

- The column and row dimensions’ results are always both presented in the main text. Their difference is however not clearly explained. E.g., what do they capture differently? Are they complementary to each other, e.g. stronger when combined? Why should we use both? Etc.
- choice_1 and choice_2 don’t seem to be explicitly defined in the paper. Do they mean when the output is positive or negative? If so, do sensory_1’s mean and sensory_2’s mean always have opposite signs? Or, do they mean when sensory_1 or sensory_2 is picked by the RNN? If so, how is the detection done? And shouldn’t context_1 always lead to choice_1 (and context_2 to choice_2) when the RNN is fully trained?

**Limitations:**

The authors have reasonably addressed the limitations of their work.

**Strengths And Weaknesses:**

Strengths
+ (Originality) The proposed tool (operative dimensions) is novel to the best of my knowledge, as well as the two main findings regarding previously unknown properties of RNNs (as summarized above).
+ (Quality) The proposed tool is well designed (as dimensions with highest impact on local dynamics when removed), easy to compute (closed-form for column dimensions), and carefully validated with lots of experiments and analyses in the main text and appendix.
+ (Clarity) The writing is overall clear and easy to follow. The figures are well-made and very helpful.

Weaknesses
- (Significance) Although the two tasks (context-dependent integration and sine wave generation) are well studied and commonly used in computational and theoretical neuroscience papers, it’s hard to predict if the proposed tool (validated on them) and results (derived from them) really generalize to harder, more realistic tasks. Additional experiment(s) using e.g. sequential MNIST, TIMIT, or any of the possible applications described in Sec 3, could greatly enhance the significance of this work.

---

> ### Author Response · Authors · 2022-08-02
> **Reply to reviewer 49zS**
>
> We thank the reviewer for the careful assessment of our work. We appreciate the positive feedback regarding operative dimensions and the suggestions for improvement. We address the specific questions and comments below:
>
> (1) _Weakness: “it’s hard to predict if the proposed tool (validated on them) and results (derived from them) really generalize to harder, more realistic tasks. Additional experiment(s) using e.g. sequential MNIST, TIMIT, or any of the possible applications described in Sec 3, could greatly enhance the significance of this work”._ \
> We thank the reviewer for this helpful suggestion. To address this point, we trained RNNs on sequential MNIST and identified the operative dimensions for them (new Supp. Fig. 15). For simplicity, we again employed vanilla RNNs, as we did for the simpler tasks used throughout the manuscript. A disadvantage of vanilla RNNs is that, as expected, they cannot approach state-of-the-art performance on sequential MNIST (Supp. Fig. 15c, f; 94% correct classification on the validation set). The advantage of using vanilla RNN is that we were able to apply the exact same methods/code as for the other tasks presented in the manuscript. The results of these analyses are mainly presented in the appendix (Section A.3.7).
> In summary, we find that our operative dimensions consistently reveal a low-d functionally relevant subspace also for sequential MNIST, as they did for the context-dependent integrator and the sine-wave generator networks (Supp. Fig. 15e-g). Compared to these simpler tasks, the networks trained on sequential MNIST require more sampling locations to obtain the operative dimensions (3950 instead of typically 120) and the functionally relevant subspace in the weight matrix is overall of higher dimensionality (58, compared to 15/29 for the context-dependent integrator/sine wave generator). Nonetheless, the relevant subspace is of substantially lower dimensionality than the full-rank weight matrix (Supp. Fig. 15b), and principal components of the weight matrix again fail to identify a similar subspace (Supp. Fig. 15c, d). Overall, these results suggest that operative dimensions will be a valuable tool in relating network structure to function also in the setting of tasks more relevant for AI.
>
> (2) _Question 1: Difference between column and row operative dimensions._ \
> Thank you for letting us elaborate on the difference between column and row operative dimensions. At present, we consider column and row operative dimensions as complementary approaches to understanding the relation between structure and function in RNN. As we now mention more clearly in the appendix (Section A.3.11), row and column dimensions describe different properties of the RNN. Column dimensions describe the “output connections” of each hidden unit, whereas row dimensions describe the “input connections”. In the current manuscript we do not exploit this difference, beyond showing that the two types of operative dimensions differ from each other (Supp. Fig. 19) and that row dimensions tend to be more (albeit weakly) aligned with the input dimensions than the column dimensions (new Supp. Fig. 11).
> From a practical point of view, the computation of column operative dimensions in vanilla RNN is more efficient than that of row operative dimensions. The reason is that only a single local column operative dimension is required to explain the recurrent dynamics at any given location in state space, a property that follows directly from the update equation of vanilla RNN (Appendix section A.2.3). In comparison, multiple local row operative dimensions are generally required at each sampling location (Supp. Fig. 20d, j).
> Ultimately, a complete understanding of the structure to function relation in RNN, and the most effective compression of the weight matrix, will likely require combining information about row and column dimensions. We agree that this would be an important direction to pursue in the future.
>
> (3) _Question 2: Definition of choice_1 and choice_2._ \
> Thank you for bringing this to our attention. We have clarified the definition of choice_1 and choice_2, and their relation to the sensory inputs and contexts, in the manuscript (Section 2). The relevant section now reads as follows:
> The network should reach choice_1 or choice_2 if the average of the contextually relevant sensory input is positive or negative, respectively.

---

> > ### Comment · Reviewer_49zS · 2022-08-09
> > **Re: Reply**
> >
> > Thank you for the additional results and clarification! I've increased my rating of the paper to 7.

---

> > > ### Author Response · Authors · 2022-08-09
> > > **Re: Reply**
> > >
> > > We thank the reviewer for considering our additional results and revised manuscript.

---

### Author Response · Authors · 2022-08-02
**General reply to reviewers**

We thank all reviewers for their valuable and interesting comments, questions and suggestions.
Besides minor changes and clarifications, we added the following main new contributions to the revised manuscript:

* Following the suggestions of several reviewers, we took first steps towards expanding our work beyond the simple, neuroscience-inspired tasks presented in the original submission. Specifically, we trained vanilla RNNs on sequential MNIST and estimated the operative dimensions for them (new Supp. Fig. 15, section A.3.7). We find that operative dimensions can identify a (relatively) low-d functional subspace of the connectivity also for this more complex task. These new experiments further demonstrate the general applicability of our proposed approach, and suggest that operative dimensions can be a useful tool to elucidate structure to function relations also in the setting of tasks more relevant for AI.
* We incorporated citations to several lines of work pointed out by the reviewers as being highly relevant to our manuscript. These additions help us to establish more connections to past work and point to future avenues to explore the implications of our findings.
* To address questions by several reviewers, we added an additional analysis to characterize the relation between our operative dimensions and other, more established dimensions describing RNN structure and function (new Supp. Fig. 10-12, section A.3.4). In particular, in the new analysis we explore relations between local operative dimensions and local linear approximations of the RNN dynamics.

---

### Meta-Review · Area_Chair_jz4x · 2022-08-24

**Recommendation:** Accept
**Confidence:** Certain

**Metareview:**

This paper defines a new way to identify a low-dimensional subspace in recurrent weight matrices of a recurrent neural network that is important for computation. All reviewers agreed the approach is novel and insightful.

**Award:**

No

---

### Decision · Program_Chairs · 2022-09-14

Accept